# Influence of data source and copula statistics on estimates of compound flood extremes in a river mouth environment

Kévin Dubois[1,2], Morten Andreas Dahl Larsen[3,4], Martin Drews[3], Erik Nilsson[1,2], Anna Rutgersson[1,2]

[1]Department of Earth Sciences, Uppsala University, Uppsala, 752 36, Sweden
[2]Centre of Natural Hazards and Disaster Science (CNDS), Uppsala University, Uppsala, 752 36, Sweden
[3]Department of Technology, Management and Economics, Technical University of Denmark, Lyngby, 2800, Denmark
[4]Danish Meteorological Office, Copenhagen, 2100, Denmark

*Correspondence to*: Kévin Dubois (kevin.dubois@geo.uu.se)

**Abstract.**

Coastal and riverine floods are major concerns worldwide as they can impact highly populated areas and result in significant economic losses. In a river mouth environment, interacting hydrological and oceanographical processes can enhance the severity of floods. The compound flood hazards from high sea levels and high river discharge are often estimated using copulas among other methods. Here, we systematically investigate the influence of different data sources coming from observations and models as well as the choice of copula on extreme water level estimates. While we focus on the river mouth at Halmstad city (Sweden), the approach presented is easily transferable to other sites. Our results show that the choice of data sources can considerably impact the results up to 10% and 15% for the river time series and 3% to 4.6% for the sea level time series under the 5- and 30-year return periods respectively. The choice of copula can also strongly influence the outcome of such analyses up to 13% and 9.5% for the 5-year and 30-year return periods. Each percentage refers to the normalized difference in return levels' results we can expect when choosing a certain copula or input dataset. The copulas found to statistically best fit our datasets are the Clayton, BB1 and Gaussian (once) ones. We also show that the compound occurrence of high sea levels and river runoff may lead to heightened flood risks as opposed to considering them as independent processes and that in the current study, this is dominated by the hydrological driver. Our findings contribute to framing existing studies, which typically only consider selected copulas and data sets, by demonstrating the importance of considering uncertainties.

## 1 Introduction

Floods can cause severe damage and disrupt activities and infrastructures in harbours and coastal communities. Flooding can result from meteorological, hydrological and oceanographic sources such as storm surges, extreme river runoff or precipitation. Storm surges correspond to seawater being pushed by the wind stress and the barometric pressure effect under deep low-pressure weather systems. Heavy precipitation can form under different conditions such as intense cyclonic activity, sometimes during the same deep low-pressure systems that cause the storm surge, or sometimes in convective weather conditions. River runoff can also have different origins, such as snow melting upstream or intense precipitation, either related

to the same large-scale weather system that could cause the storm surge or separately. Hence, several processes could contribute to compound effects, but also independently cause damage and disruption to activities in the coastal zone. River runoff and precipitation may take some time to drain into the sea, and the flow from land to the sea can therefore be slowed down or even momentarily become blocked when storm surges happen (Wahl et al., 2015). This process can be referred as coastal backwater effects while the water level at the river mouth increases due to high river discharge or high sea level or compounding effects (Feng et al., 2022).

Settlements and infrastructure located in river mouth environments are inherently susceptible to all of the above. The combination of multiple factors, extreme or not, happening at the same place simultaneously, successively, or consecutively, can potentially lead to larger, compounded floods, and more severe impacts on the environment and society. Compound flooding can also result when preceding conditions amplify the impact of the event (Andrée et al., 2023; Zscheischler et al., 2020; AghaKouchak et al., 2020). Even if trends over the last forty to sixty years are estimated with high uncertainties, it is likely that extremes including compound events are becoming more severe in Northern Europe in a changing climate (Rutgersson et al., 2021).

Couasnon et al. (2020) highlight the importance of considering interactions referred to their co-occurrence probabilities between river discharge and storm surge extremes in river mouth environments. They demonstrate that dependencies between these drivers are not random and may result from relations between weather systems at the synoptic scale with local conditions such as the topography. Ward et al. (2018) study the dependence between river discharge and skew surge at the global scale where significant dependency is found in several stations in Europe, mainly located around the UK coastline. Hendry et al. (2019) also highlighted those dependencies around the UK coast and linked the spatial variability found with differences in storm characteristics. In Northwestern Europe, it has been shown that the fluvial flood hazard increases with high sea levels and stronger storms, and that this may be critical in populated and low-elevation coastal areas (Ganguli and Merz, 2019). Increasing trends within the last decades in the magnitude and frequency of coastal compound floods between river discharge and sea levels are found for gauges between 47°N and 60°N latitude while decreasing trends are highlighted for gauges > 60°N in Northwestern Europe (Ganguli and Merz, 2019), where rare occurrences of compound floods are reported due to a decrease in relative sea level rise across Nordic countries due to vertical crustal movement (Weisse et al., 2021). For example, Eilander et al. (2020) find the Baltic Sea and the Kattegat basin to be particularly susceptible to compound flood hazards based on the dependency between skew surge levels and river discharge. Without considering the occurrence of storm surges, Eilander et al. (2020) further show that flood depths are underestimated and subsequently so is the estimated number of people exposed to river floods in this area. Meanwhile, Moftakhari et al. (2017) demonstrate that sea level rise (SLR) is likely to increase the impacts from compound flooding by 2030 and 2050 under the Representative Concentration Pathways (RCPs) 4.5 and 8.5 for eight major cities around the US coastline.

Compound flooding in coastal areas can also be caused by a combination of heavy precipitation inducing large runoff and high sea levels (Bevacqua et al., 2019). Hence, the probability of compound flooding is expected to significantly increase in the Baltic and North Sea areas, where an event with a current return period (RP) of around sixty years is projected to occur

every ten years in 2100 due to the combination of SLR and increased extreme precipitation (Bevacqua et al., 2019). However, Ganguli et al. (2020), in a coupled statistical-hydrodynamic modelling framework showed that projected changes in compound flood hazard are limited to 34% of the sites with a substantial role of SLR in modulating compound flood hazard in Northwestern Europe.

Not taking compound flooding effects into account may result in an underestimation of the flood hazards in the coastal zone, including river mouths (Ward et al., 2018). Thus, analyzing and understanding these events is of high relevance to coastal communities. In this study, we evaluate the potential impact of extreme hydrological and oceanographic coastal events on the coastal city of Halmstad (Sweden), which is a port, industrial, and recreational city at the mouth of the Nissan river. Halmstad is located on the west coast of Sweden (fig. 1) and has been chosen as it is naturally prone to coastal, fluvial, and pluvial flooding. The area is subject to extratropical cyclones (Hoskins and Hodges, 2002; Dacre et al., 2012), resulting in rather high sea levels by storm surges for the area (Wolski et al., 2014). According to the Swedish Meteorological and Hydrological Institute (SMHI), Halmstad recorded the highest ever sea level measured in Sweden of 235 cm on the 29th of November 2015 during the wind storm "Gorm". Halmstad has also been severely impacted by river floods. While Nissan is the main river crossing the city of Halmstad, smaller rivers are also present and can create floods, such as Fylleån's river. Finally, the West Coast of Sweden is found to be one of the areas in Sweden expecting the most significant impacts due to SLR during this century (Hieronymus and Kalén, 2020). Thus, nearby studies have stressed the necessity to update coastal protection measures along the Swedish (Hieronymus and Kalén, 2022) and the German Baltic Sea shorelines (Kiesel et al., 2023). To help guiding and communicating with the local municipalities about their continued work to protect coastal areas from flooding we considered it useful to pick one site in this area as an example to showcase the applied methods and their results.

The main goal of the current study is to investigate the impact of different data sources, methodology, and representation of compound extremes on estimates of extreme water levels. Our main focus is to evaluate the sensitivity to data sources of compound flood hazards from river discharge and sea level, potential sources of uncertainties. Using Halmstad as an example, we explore the potential influence on flood hazard assessments related to compound effects from river flooding within the coastal area.

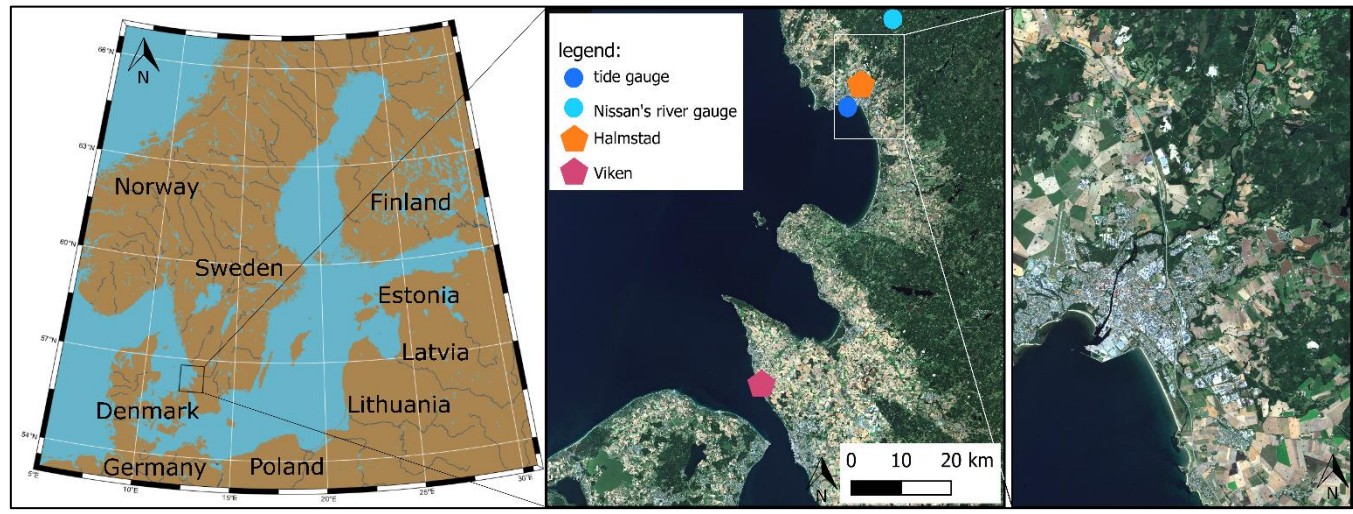

**Figure 1: Regional map of southern and central Sweden with a zoom around our study area and the city of Halmstad (Pawlowicz, 2020; Sentinel 2 cloudless, 2021)**

## 2 Data and Methods

In the following, compound effects are defined in terms of the co-occurrence probabilities between coastal sea level and river discharge when at least one of the two is subject to an "extreme" value. An event is considered in the extreme range when a studied variable reaches its annual maximum value. The annual maximum values will differ between different years in a range between 84 to 235 cm for sea level and 88 to 271 $m^3 s^{-1}$ for river discharge. The correlation between co-occurring events has been studied as it provides insight into the relationship between each set of two variables. The exceedance probability of getting an extreme river discharge associated with a high sea level and the opposite permits the assessment of the potential compound effects between those two processes but it does not determine impacts from compound flooding either in terms of estimating water level or computing inundation depths. Hybrid statistical-hydraulic modeling frameworks have been introduced to answer such issue and study compound flood impacts (Jane et al., 2022; Moftakhari et al., 2019; Gori et al., 2020; Olbert et al., 2023).

Figure 2 presents the main steps of the workflow describing the methodology; this is described in the following sub-sections. Firstly, we analyzed different time-series records of sea level and Nissan River discharge data from models and observations at Halmstad using extreme value theory and a Generalized Extreme Value (GEV) distribution (Coles, 2001) – hereby referred to as the "univariate approach" - to estimate Return Levels (RLs) on every driver independently. Secondly, we defined sets of coupled events based on single variables. Thirdly, we analyzed the correlation between sea level and river discharge events. If this analysis indicated potential for compound events, we studied the co-dependency between the two variables by fitting a copula distribution function (Sadegh et al., 2018). We finally performed a statistical analysis of the compound events to study the differences between each data source and its associated uncertainties.

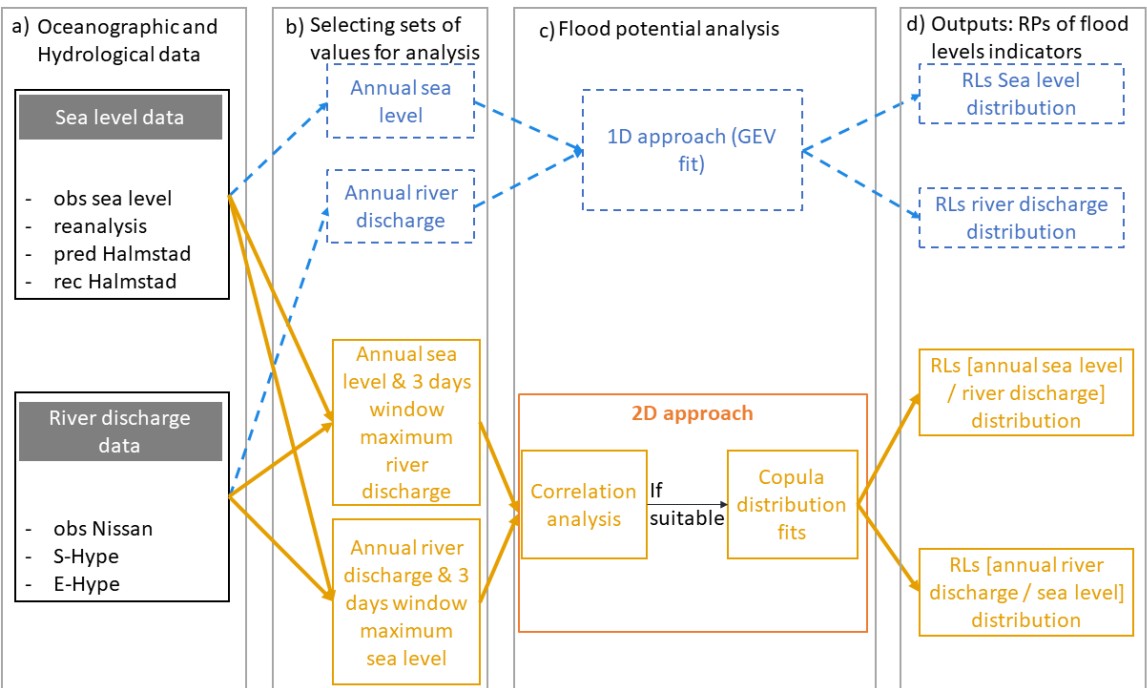

**Figure 2: Workflow describing the methodology used in this paper, starting from the oceanographic and hydrological data (a) to the univariate (in blue and dashed arrows) and bivariate (in orange and continuous arrows) approaches used for flood hazards analysis. Panel b represents the selection sets, the analysis is then described (c) and the analyses' results are issued (d).**

## 2.1 Data

An analysis of time-series records of sea level and river discharge at Halmstad was carried out. As mentioned above, a univariate distribution was initially fitted based on the GEV distribution and extreme value theory (Coles, 2001) for each time series collected. The temporal differences in the lengths of each dataset induce substantial differences and associated uncertainties, which dominate in the case of extreme RP. Consequently, a moderately extreme 30-year RP event was chosen as the maximum value considered. For comparison, we also consider more frequent events with a 5-year RP.

### 2.1.1 Sea level data

Figure 3 displays the different sea level datasets used (fig. 3-a) and their corresponding univariate extreme value analysis (fig. 3-b).

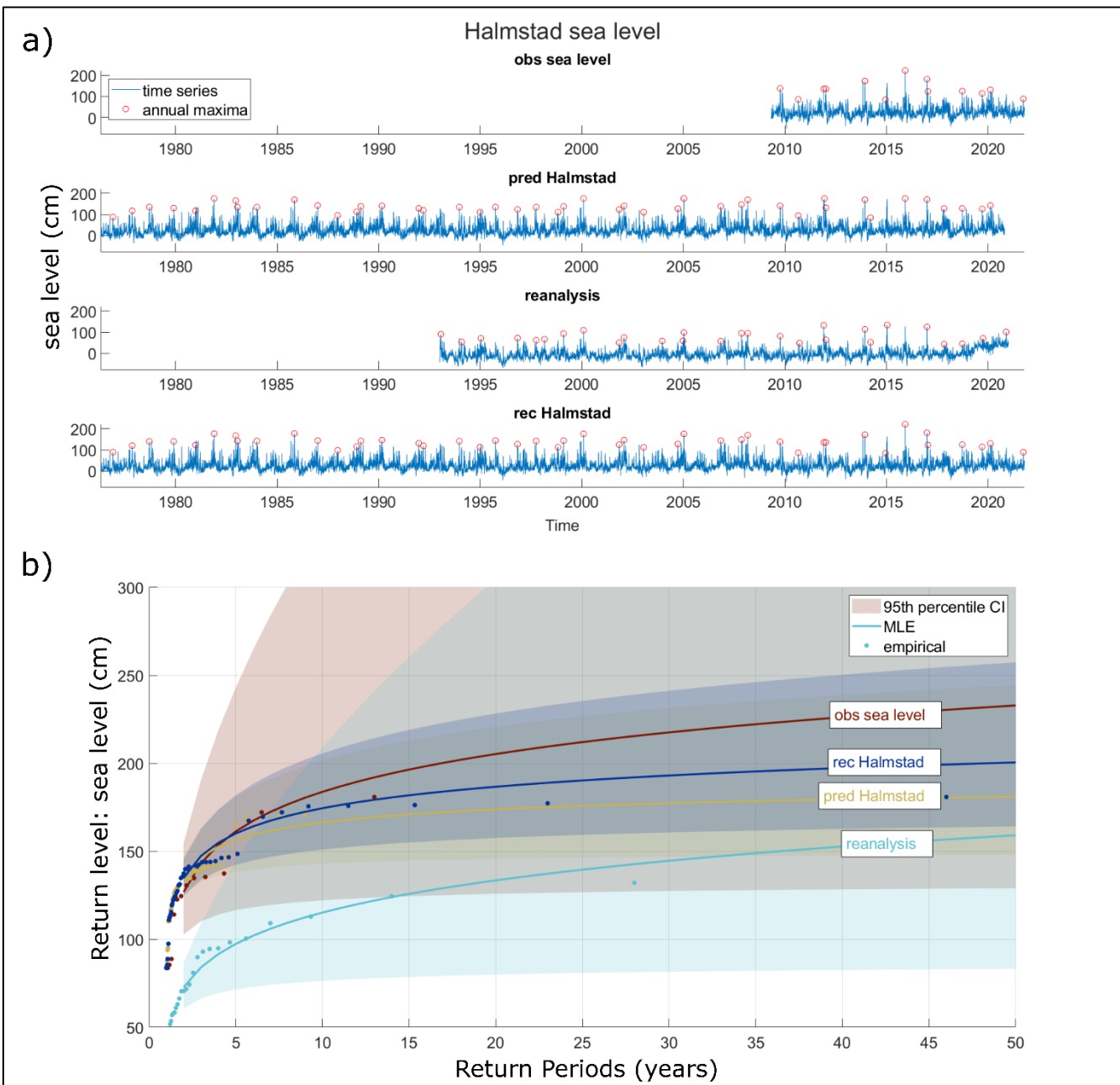

**Figure 3: Halmstad sea level time series and annual maxima from different sources: observations ("obs sea level"), reconstructed ("rec Halmstad") and predicted ("pred Halmstad") derived from a machine learning model trained on data from the Viken station, and reanalysis ("reanalysis") datasets (panel a). RLs estimated from corresponding GEV fits of each dataset and the associated 95th percentile confidence intervals (background colours). The dots depict empirical data (panel b).**

- *Observations*

Hourly sea level observations data were obtained from SMHI at Halmstad's tide gauge denoted as the station "HALMSTAD SJÖV" with station number 35115 in the open database provided by SMHI (fig. 1). This hourly sea level time series is transformed to a daily time series using the maximum hourly data within the day. However, to carry out this analysis, the period with sea level observations (titled "obs sea level") was insufficient as only 13 years (from 2009 to 2021) are available. To extend this sea level record, a set of reanalysis data and a machine learning approach have been investigated and used.

- *Reanalysis*

Hourly sea surface variations (in m) covering the period from 1993 to 2020 with a spatial resolution of approximately two nautical miles have been provided by the Copernicus Marine Environment Monitoring Service's (CMEMS) Baltic Monitoring and Forecasting Centre (BAL MFC) (CMEMS, 2022). This reanalysis uses the ice-ocean model NEMO-NORDIC (Pemberton et al., 2017), and the data are assimilated with the Localized Singular Evolutive Interpolated Kalman (LSEIK) method (Nerger et al., 2005). Data is extracted from the closest grid point to Halmstad's tide gauge and the hourly data was changed to a time series of daily maxima for our purpose to focus on extremes. This sea level dataset is named: "reanalysis".

- *Machine learning model*

A probabilistic machine learning method, Random Forest (RF), is used (Breiman, 2001). Sea level records from the neighbouring station of Viken (station named "VIKEN", number 2228 in the SMHI database) is used to train the RF model over an eight year-period, where it is correlated with Halmstad's observed sea level. The resulting sea level estimates at Halmstad include both mean predictions and standard deviation to assess uncertainties and variability following the methodology introduced by Dubois et al. (2024). The last three years of available data at Halmstad are used to validate the RF model, emphasizing extreme events predictions. The RF model is used to produce a first dataset called: "predicted Halmstad" ("pred Halmstad") and a second one named: "reconstructed Halmstad" ("rec Halmstad").

The predicted Halmstad dataset provides daily sea level (in cm) for the full period of available sea level observations from the station at Viken, here from 1977 to 2021.

The reconstructed Halmstad dataset provides daily sea level (in cm). It joins both sets, i.e., combines observations from Halmstad from 2009 to 2021 and the predicted Halmstad data from the RF model from 1977 to 2009. Thus, it also covers the period from 1977 to 2021. Further attempts to enrich the RF model by including reanalysis data (i.e., as part of the training) did not improve the predicted sea levels in the reconstructed data sets significantly, which emphasized the need for local observations. These were fortunately available, even if not for the entire more extended period. Similar findings, that is, significant improvements when using local observations as means to train a machine learning of sea level were previously found in this region (e.g. Hieronymus et al., 2019).

It is not only the length of the observation period that is short. Also, the reanalysis dataset exhibits a bias and does not predict the observed extreme sea levels. Accordingly, the uncertainty estimated from both univariate GEV analyses are large (fig. 3-b). The predicted and reconstructed data sets yields result with a smaller uncertainty range. Hence, the reconstructed data set, which is based on observations when available, was chosen as the best source of sea level information for the bivariate analysis.

### 2.1.2 River runoff data

Figure 4 presents the different river discharge datasets obtained (fig. 4-a) and their corresponding univariate extreme value analysis (fig. 4-b).

- ***Observations***

River discharge data were obtained from SMHI at station 2471: "Nissaström" (fig. 1), covering a basin of 2437 km$^2$. Observations of daily river discharge in m$^3$ s$^{-1}$ (obs Nissan) have been provided from 1997 to 2021.

- ***E-Hype model***

Modelled river discharge data are taken from the Hydrological Predictions for the Environment (HYPE) model, which simulates water flows and quality at different spatial scales (Lindström et al., 2010), a model detailed description can be found at http://www.smhi.net/hype/wiki/doku.php?id=start. Daily temperature and precipitation values are used as dynamic forcing in this model. The European HYPE model: "E-Hype2016_version_16_g" (E-Hype) provides daily river discharge (in m$^3$ s$^{-1}$) from 1989 to 2021 (https://vattenwebb.smhi.se/om-vattenwebb). The model performs better for annual and seasonal flows compared with daily and extreme flows (Donnelly et al., 2016).

- ***S-Hype model***

SMHI has set up the HYPE model for Sweden, now used operationally to forecast hydrological conditions over Sweden, such as floods and droughts. It covers all of Sweden (450000 km$^2$), where the country has been divided into sub-basins of 28 km$^2$ on average (Strömqvist et al., 2012). S-Hype3 model's data (S-Hype) of daily river discharge (in m$^3$ s$^{-1}$) has been provided from 2004 to 2020 (Donnelly et al., 2016). The model seems to slightly underestimate high flow peaks with high flow statistics differing by around +-10% whereas the mean flow is highly reliable (Bergstrand et al., 2014).

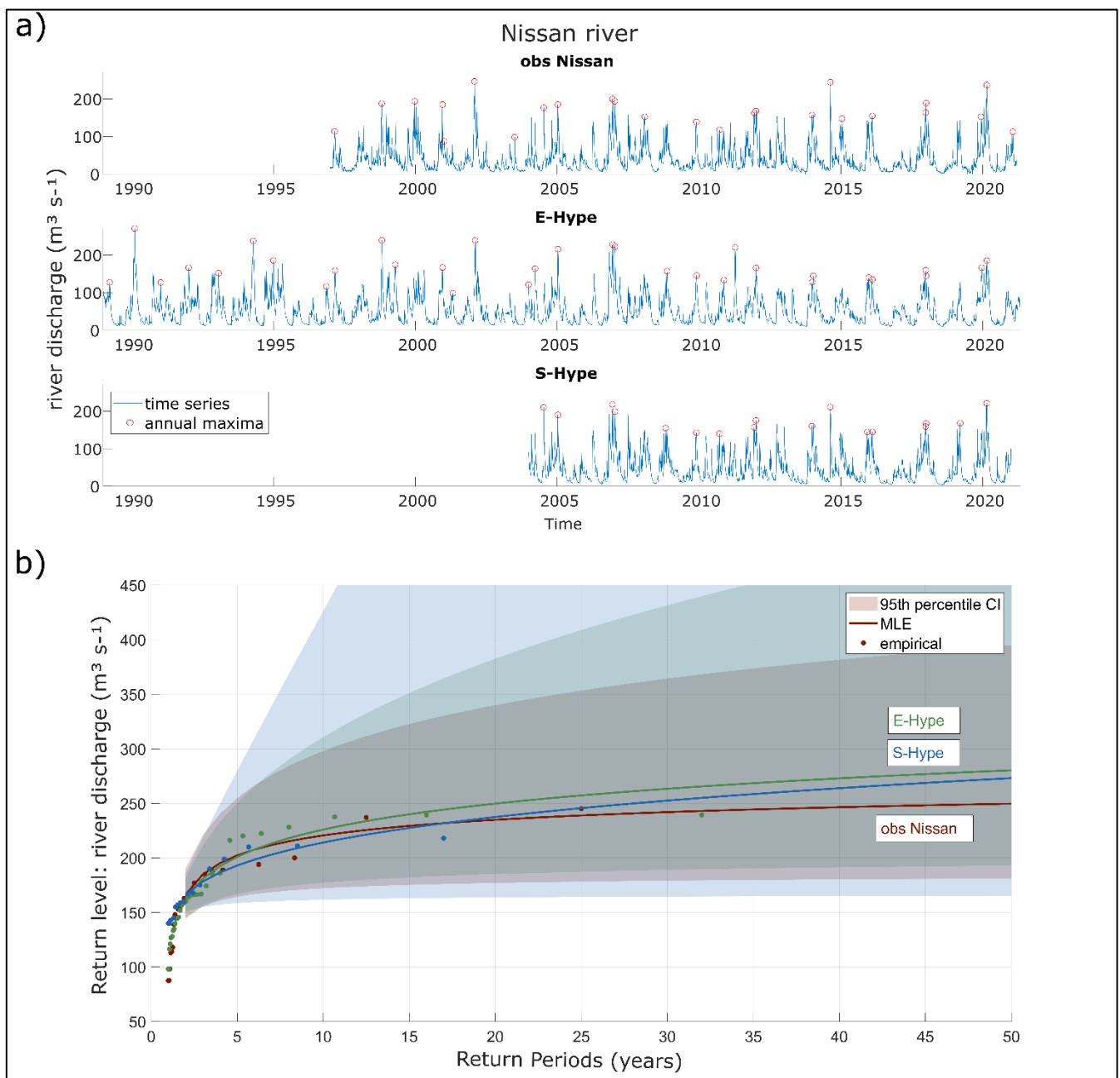

**Figure 4: Nissan's river time series and annual maxima from different sources: observations ("obs Nissan"), European HYPE model ("E-Hype") and Swedish HYPE model ("S-Hype") from SMHI (panel a). RLs derived from GEV fits to each dataset are shown with 95th percentile confidence intervals (background colours). The dots are the empirical data (panel b).**

The available time series associated with the S-Hype model is rather limited, leading to a wide uncertainty band when carrying out the univariate analysis (fig. 4). Conversely, the Nissan observations and E-Hype data sets lead to RLs that are associated

with more bounded uncertainty estimates. In this light, we choose the E-Hype dataset for the bivariate analyses as the data are available over a more extended period. The largest RLs are seen for the E-Hype dataset for RPs above five years.

### 2.1.3 Sets of coupled events

To study compound events in a river mouth environment from both a river discharge and sea level perspective, we defined two different sets of events based on the data discussed above. The first one paired sea-level annual maxima ($S_n$) and associated daily maximum river discharge ($q_n$) within a defined time period centered on the date of $S_n$ (+/- $\Delta$ days). The second one pairs river discharge annual maxima ($Q_n$) and associated hourly maximum sea levels ($s_n$) within a defined three days window centered on the date of $Q_n$ (+/- one day) (Couasnon et al., 2020; Moftakhari et al., 2017; Sadegh et al., 2018). Each of

the four sea level time-series observed and modelled records were then correlated with each of the three river discharge ones, which makes up a total of twelve different datasets (table A2).

### 2.2 Statistical analysis

- *Univariate analysis*

To estimate the extreme values of Nissan's river runoff and Halmstad sea levels, and their associated RPs, a GEV distribution was fitted to the annual extremes separately for each time series record (Coles, 2001; Ahsanullah, 2016). This was done using the MATLAB-based GEV-fitting algorithm, which provides parameter estimates and 95% confidence bands. Here, we do not make any assumption concerning the dependence between the two variables of interest, sea level and river discharge; each variable is modelled independently based on its own marginal distribution.

- *Bivariate analysis*

Initially, the Pearson, Spearman and Kendall's correlation coefficients and the associated p-values were calculated for each of the twelve collated data sets to assess whether there was a relationship between river runoff and sea level. The usual threshold value of 5% was defined as evidence for rejecting the H0 null hypothesis, that is, the two variables are independent. When p-

values were found to be lower than the threshold, the null hypothesis could be rejected, and the two variables similarly found to show significant dependency. However, when p-values are above 5%, H0 cannot be rejected, so the two variables can be independent.

To represent the compound extremes, we apply copula modelling, which has been found to be useful for representing a joint probability (Hao et al., 2016). The analyses were carried out using the Multihazard Scenario Analysis Toolbox (MhAST),

Version 2.0 from Sadegh et al., (2018). The copula method models the dependence structure of the two random variables (Joe, 2014; Sadegh et al., 2017). It links or joins individual univariate distributions into a joint multivariate distribution that has a specified correlation structure (Tootoonchi et al., 2022). The MhAST toolbox fits 25 different copulas to an input dataset. It

first calculates the best possible fitting marginal distribution for each univariate dataset. It then proposes the best copula fit based on the Maximum Likelihood, Akaike Information Criterion (AIC), and the Bayesian Information Criterion (BIC). The root mean square error (RMSE) and Nash-Sutcliffe efficiency (NSE) values are calculated for each copula. Here, we evaluated the difference between each of the copula fits. Joint RP can then be calculated based on the copula but results in statistically similar infinite combinations of sea level and river discharge values for each RP event (Sadegh et al., 2018). The scenario with the highest density along the closed-form joint probability density function of the copula permits the identification of the "most likely scenario". This scenario is based on the copula fit parameters, which represent the statistical relationships between the individual hazard components coming from the input samples. An uncertainty analysis was also carried out using the MhAST toolbox with a "Weighted Average" and a "Maximum Density" approach (Sadegh et al., 2018). This first approach reproduces a distribution of potentially compound hazards. Based on the determined joint probability contours, random samples are weighted from the critical joint RP. 1000 weighted samples are randomly drawn from it. Therefore, a sample with a higher joint probability density value has a higher chance of selection. This approach effectively generates a distribution of potential compound hazards while considering the underlying copula structure. This provides a comprehensive overview of the overall range of possible compound hazards. The second approach is based on the "most likely scenario" and provides an uncertainty range around it. A range of possible most likely scenarios can be generated based on the different copulas issued from the same copula family that best describes the input datasets, allowing for the quantification of uncertainties around this central scenario (Sadegh et al., 2018).

Two types of Hazard Scenarios (HSs) have commonly been proposed to study the hazard of compound floods related to sea level and river discharge (Salvadori et al., 2016; Moftakhari et al., 2019; Serinaldi, 2015). The "AND scenario" corresponds to a scenario where both the river discharge and the sea level are large enough to make a bivariate occurrence hazardous meaning that both high sea levels and river discharge exceed the respective random variables concurrently. The "OR scenario" corresponds to a scenario where either the river discharge or the sea level or both are large enough to make a bivariate occurrence troublesome meaning that either of the extremes exceeds the respective random variable with a time offset within a limited time interval (Requena et al., 2013).

### 2.3 Methodology

Firstly, a correlation analysis was carried out for each set, as proposed in section 2.2. This analysis investigated the significance of independence between the sea level and river discharge during extreme occurrences. Then, each set was used as input to the MhAST toolbox, which performed the compound analysis and returned 25 copula fits ranked depending on different criteria (section 2.2). Among the 25 copulas fitted, only the ones presenting a closed-form joint probability density function (Sadegh et al., 2017) were further investigated since, in these cases, "most likely scenarios" and their associated uncertainties can be defined. Chosen RLs were calculated for each copula, and their uncertainties were assessed. Adopting the "AND scenario" (see above) permitted us to investigate the hazard of compound events only, highlighting the dependency between sea level and river discharge during extreme events. Conversely, the "OR scenario" was finally preferred when

looking at RLs as this looks into the "total" flood hazard, whether originating from hydrological, coastal sources or both in combination.

To compare and evaluate the role of copulas and the role played by sea level and river discharge, respectively, a notion of normalized difference value (NDV) was introduced. We defined it as the normalized difference between the RL values of interest from the bivariate analyses. Here, we normalize relative to the corresponding E-Hype univariate RL, which yields a representative dimensionless quantity. It should be pointed out that this quantity does not represent the "amplification" with respect to the univariate case since, as shown by Serinaldi (2015), one cannot compare RLs of different dimensionality. Suppose we investigate the resulting spread from using different copulas (based on the most likely scenarios within one set under the 5-year RP); the NDV is calculated as the difference between the maximum and minimum value of the most likely scenarios for any copula within a specific set divided by univariate 5-years RL derived from the E-Hype data set (section 3.2). When we look into the sensitivity of river discharge datasets, the sea level dataset is fixed, and the NDV is measured as the normalized difference between the maximum and minimum values of the most likely scenarios of best fits among the three sets of associated river discharge divided by the corresponding E-Hype univariate RL; and vice-versa, when looking into the sensitivity of sea level datasets. The NDV term indicates the magnitude of change or difference in RL results we can expect when choosing a certain copula or input dataset. Very small NDVs suggest that the corresponding choice of a variable of interest does not strongly influence the results. In contrast, large NDVs indicate that a particular choice results in significant differences.

## 3 Results and discussion

From the rank correlation analysis, the datasets based on sea level annual maxima ($S_n$, $q_n$) did not reveal any significant dependency (i.e., "compoundness") between sea level and river discharge, and therefore, no copula analysis was done (table A.1). In this case, the univariate analysis seemed to fit best under the proposed conditions of this study. Conversely, the datasets based on river discharge annual maxima ($Q_n$, $s_n$) yielded significant dependencies, suggesting a possible compound impact on river discharge. In subsections 3.1 and 3.3.1, we look into the "AND scenario" as we investigate the compound hazard only. In the subsections 3.2 and 3.3.2, we mainly focus on the "OR scenario" (see above) as we are interested in the total flood hazard driven regardless of the situation (oceanographic, hydrological or compound). The set rec Halmstad / E-Hype is chosen as our base case because it has the longest co-occurring period (table A.2).

### 3.1 Dependency / Independency of the variables

Figure 5 and Table A1 show the dependency between the river discharge annual maxima and associated sea level local maxima ($Q_n$, $s_n$) event sets as expressed in section 2.1. Figure 5 displays the best copula distribution fit: BB1 from the rec Halmstad / E-Hype set under the "AND scenario" hypothesis for the 2, 5, 15 and 30-year RP. The full lines depict the RLs considering sea level and river discharge as dependent variables (derived from the best copula distribution fit). In contrast, the

dashed lines show analogous results when assuming the two variables to be independent. The figure shows that the lines do not overlap, highlighting a dependency between both variables. Also, for all RLs presented, each RL from the independent hypothesis (dashed line) is placed below each corresponding RL from the dependent hypothesis (full line), supporting the

300 hypothesis that compound events lead to higher flood risks when considering compound extremes as also found in Bevacqua et al. (2017) where they studied compound hydrological and oceanographic floods in Ravenna (Italy). Therefore, for example, a 30-year RP, when looking at the independent variables, would become a 13-year RP when considering the variables' dependency. This frequency increase comes from the compound effects and can be highlighted for each RP and copulas tested. Also, the dependency between extreme hydrological conditions and high oceanographic ones stresses the presence of

305 compound effects, which lead to higher levels of river discharge and sea level during such events at the estuary. Joint probability contours are derived, which permits obtaining a probability of co-occurrences for a possible event along each curve, which is later used to carry out the uncertainty analysis (Sadegh et al., 2018). For example, along the 5-year RP curve, the probability of getting a 5-year RL of 180 $m^3 s^{-1}$ river discharge and 93 cm sea level is higher than getting one of 201 $m^3 s^{-1}$ and 20 cm or one of 101 $m^3 s^{-1}$ and a 112 cm one (fig. A1).

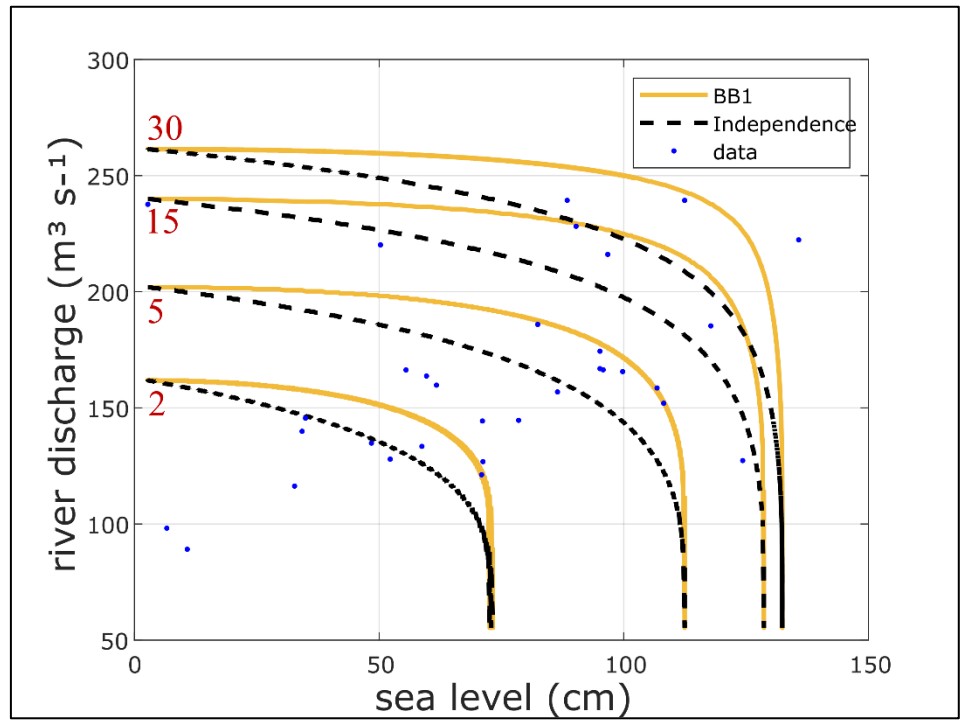

**Figure 5: RLs for base case set rec Halmstad / E-Hype. Full lines correspond to the return period (RP) isolines for joint probability (AND scenario) of river discharge (y-axis) annual maxima and associated sea level (x-axis) maxima ($Q_n$, $s_n$). The dashed lines represent the distribution fit, assuming the independence between the variables. Blue dots show observed data. BB1 copula is used**

**to model the dependence of river discharge annual maxima and associated sea level maxima calculated for each RPs visible in red text (2, 5, 15 and 30 years).**

### 3.2 Compound hazard potential on river floods

The focus is on river discharge RLs as a proxy for fluvial flooding indicators. Figure 6 represents the 5- and 30-year
river discharge RLs from the set rec Halmstad / E-Hype under the "OR scenario" hypothesis for each copula tested and its
associated uncertainties values from two approaches. The best copula fit selected based on the different criteria as AIC in this
case (section 2.2) is BB1 (red diamond). The stars and diamonds represent the maximum density of the calculated RL for each
copula, which can be interpreted as the most likely scenario under the bivariate analysis.

- ***RP = 5 years***

The 5-year RL from the E-Hype model is 201 $m^3 s^{-1}$ with a 95th percentile confidence interval of 167-246 $m^3 s^{-1}$ under the
univariate GEV distribution fit. The BB1 copula fit has a 5-year RL "most likely scenario" of 220 $m^3 s^{-1}$. Among all tested
copulas, their 5-year RLs of "most likely scenarios" differ around 26 $m^3 s^{-1}$, all between 208 and 234 $m^3 s^{-1}$. The RL copulas'
uncertainties are displayed with the boxplots from two methods: the "Weighted Average" approach showed with the outlined
error bars, and the "Maximum Density" approach showed with the filled error bars. The "Weighted Average" approach gives
more considerate uncertainty ranges than the "Maximum Density" (section 2.2). Indeed, for the best copula fit, the "Maximum
Density" approach looking at the uncertainty of the "most likely scenarios" results in a narrow band of a maximum of 19 $m^3$
$s^{-1}$ per copula against a more extensive range of 159 $m^3 s^{-1}$ going from 202 to 361 $m^3 s^{-1}$ with the "Weighted Average" approach.
All copulas present a similar pattern.

Moreover, the RL uncertainties for the "Maximum Density" approach are all located within the 95th confidence interval of the
univariate RL. However, the "Weighted Average" approach gives a 75th percentile of around 255 to 269 $m^3 s^{-1}$ and a nonoutlier
maximum of around 324 to 361 $m^3 s^{-1}$ above the 246 $m^3 s^{-1}$ corresponding to the 95th percentile of the univariate GEV fit,
indicating the importance of considering bivariate analysis method. The BB1 copula chosen as the best fit here by the different
evaluation criteria mentioned in section 2.2 presents neither the smallest nor largest uncertainty band.

- ***RP = 30 years***

The 30-year RL from the E-Hype model is 263 $m^3 s^{-1}$ with a 95th percentile confidence interval going from 189 to 431 $m^3 s^{-1}$
under the univariate GEV distribution fit. The BB1 copula fit has a 30-year RL of 278 $m^3 s^{-1}$. The copulas' 30-year RL of
"most likely scenarios" differ by around 25 $m^3 s^{-1}$, with all of them between 267 and 292 $m^3 s^{-1}$, except for the Gaussian copula.
For all copulas except the Gaussian one, the "Weighted Average" approach gives a more extensive uncertainty range than the
"Maximum Density" one. Indeed, for the best copula fit, the "Maximum Density" approach results in a relatively narrow band
of 24 $m^3 s^{-1}$, going from 272 to 296 $m^3 s^{-1}$, against a more extensive range of 92 $m^3 s^{-1}$ going from 261 to 353 $m^3 s^{-1}$ with the
"Weighted Average" one. All copulas except the Gaussian and the Tawn ones present a similar pattern. Moreover, all RL

uncertainties for both uncertainty analysis approaches are within the 95[th] confidence interval of the univariate RL for the 30-
350  year RP.

- *Sensitivity to the choice of copula*

For both 5- and 30-year RPs, the copulas and their associated uncertainties present a similar pattern. Depending on the choice

of copulas, the most likely scenarios differ up to 26 $m^3 s^{-1}$ for the 5-year RP and up to 25 $m^3 s^{-1}$ for the 30-year RP, with the

Tawn copula giving the minimum value and the Fischer-Kock and FGM copulas giving the maximum value. When only

looking at the most likely scenarios values for each copula, they differ in a range approximately equal to 13% and 9.5% for

the 5- and 30-year RPs, respectively (table A3). For each copula, the uncertainties' relative errors based on the "Maximum

Density" approach differ from 1% (Gaussian) to 9.9% (Fischer-Kock) and from 2.3% (Joe) to 52% (Gaussian) for the 5- and

30-year RPs respectively; from 63% (Joe) to 75% (Fischer-Kock) and from 16% (Joe) to 35% (Fischer-Kock) for the

"Weighted Average" approach. For comparison, the relative errors for the univariate GEV fit are around 41% and 92% for the

5- and 30-year RPs, respectively. When considering the "Maximum Density" uncertainties, all RLs of all copula are in the

range of 208-234 $m^3 s^{-1}$ for the 5-year RL and 262-398 $m^3 s^{-1}$ for the 30-year RL (Table A3; fig. 6).

These differences in resulting RLs emphasize the importance of the role played by the choice of copulas and the consideration

of quantifying uncertainties.

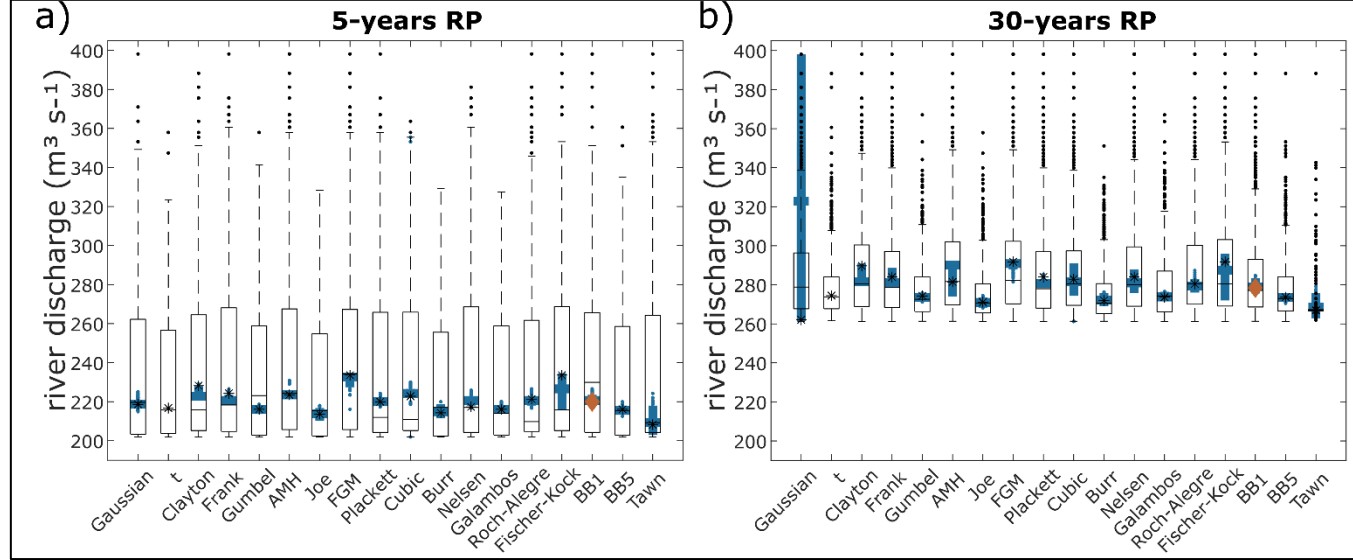

**Figure 6: Fluvial component in bivariate events with 5- (a) and 30-year (b) Return periods from copula fits for rec Halmstad / E-Hype. Each column represents a copula distribution fit. Stars represent the most likely scenario return values from each copula for each set, and the red diamond is the best copula fit. Two uncertainty approaches are displayed as boxplots, giving a statistical**
**summary. Median, first, and third quartiles are represented in each box, whiskers represent minimum/maximum values, and dots represent outliers. Outlined boxplots correspond to the "Weighted Average" approach, and filled ones to the "Maximum Density" approach.**

**3.3 Sensitivity analysis on compound flood hazard potential -OR scenario-**

This section focuses on the impact of data sources on resulting RP statistics, aiming to compare copula analyses considering compound events. As seen in section 2.1, we have twelve possible data sets to analyze for Halmstad city extracted from models and observations. As mentioned in section 2.1, the univariate analysis presents different results, including RL values and confidence intervals for each river runoff time series.

5-year and 30-year univariate RLs of river runoff, respectively, differ by around 9 $m^3 s^{-1}$ and 21 $m^3 s^{-1}$ with values of 202 $m^3 s^{-1}$
and 241 $m^3 s^{-1}$ based on observation gauge (red); 193 $m^3 s^{-1}$ and 252 $m^3 s^{-1}$ based on S-Hype model (blue); 201 $m^3 s^{-1}$ and 263 $m^3 s^{-1}$ based on E-Hype model (green) as displayed in Figure 4-b. However, uncertainties associated with the 95[th] percentile confidence interval differ vastly from respectively around 86 $m^3 s^{-1}$ and 185 $m^3 s^{-1}$ (observation); 121 $m^3 s^{-1}$ and 811 $m^3 s^{-1}$ (S-Hype); 79 $m^3 s^{-1}$ and 242 $m^3 s^{-1}$ (E-Hype) as displayed with the background colours on the figure.

**3.3.1 Dependency / Independency of the variables**

Figure 7 presents resulting RLs for combined ranges of each variable set for the 5- and 30-year RPs as in Figure 5, but with results from six different data sources to study the resulting impacts. The dependency is evident for each set, with each full line moved away from its corresponding dashed line, highlighting the dependency and compound effects for any sets tested. The differences between solid and dashed lines in Figure 7 are typically contained within about 20 cm sea level or 25 $m^3 s^{-1}$ river discharge based on the maximum distance between the copula and independence cases on rays coming from the
origin; constituting about 10-15% of the extreme 5- and 30-year RLs for the site with a gap increasing with higher RPs. At first glance, these differences may be perceived as a reasonably small compound effect, but every little increase in extreme situations can have a consequence for society. It should be noted that switching data sources may have a significant effect on estimated RLs; hence, both method and choice of data are essential.

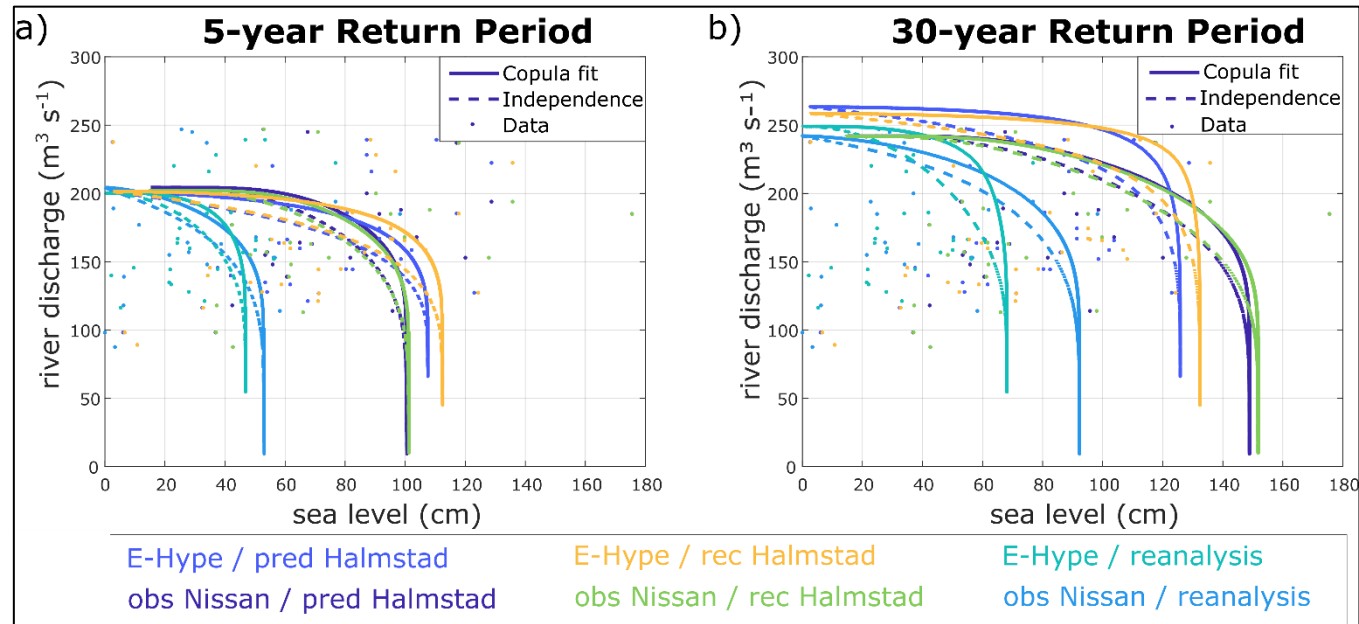

**Figure 7: 5- (a) and 30-year (b) Return periods isolines for joint probability (AND scenario) of river discharge (y-axis) annual maxima and associated sea level (x-axis) maxima ($Q_n$, $s_n$) for Halmstad. Full lines implement the compound effect, and dashed lines represent fit, assuming independence between both variables. Dots show observed data. The best copula fit is used to model the dependence of ($Q_n$, $s_n$) calculated for each set visible in coloured text.**

Some sets behave similarly as their corresponding dashed and full lines almost overlap as for the sets obs Nissan / pred Halmstad & obs Nissan / rec Halmstad or E-Hype / pred Halmstad & E-Hype / rec Halmstad in both 5- and 30-year RPs (fig. 7). This similarity emphasizes that river discharge dominates over sea-level inputs the co-occurrence probabilities of bivariate hazardous events.

## 3.3.2 Compound hazard potential on river floods

The most likely scenarios of 5- and 30-year RPs and their associated uncertainties on the different sets are calculated as described in section 2.1. This study focuses on extreme hydrological events associated with oceanographic conditions and, therefore, concentrates on the RLs of river discharge. Figures A2 and A3 display those results for each set in the same way as Figure 6. Figure A2 returns the results of the 5-year RP and Figure A3, the 30-year RP analysis.

Under the 5-year RP, lower RLs are found for all obs sea level sets (fig. A2), corresponding to the data available and short duration of overlapping periods, with a maximum of 13 years and limited by the extent of sea level observations. Under the 30-year RP, lower RLs are found for the set E-Hype / obs sea level and all S-Hype sets except for S-Hype / obs sea level, which presents the most extreme values (fig. A3).

For the 5- and 30-year RPs, the three sets associated with the E-Hype model, which show statistical significance, lead to similar and higher values, respectively, than all the other sets. The last set showing statistical significance is associated with

the S-Hype model and leads to somewhat different results between the 5-year RP, with slightly lower RLs, and the 30-year RP, with generally higher RLs (figs. A2 and A3). It stresses that the dependence changes the RP results as also shown by Santos et al. (2021) which studied compound surge and precipitation events in a case study in the Netherlands.

All most likely scenario values calculated from the copula analysis under both the 5- and 30-year RPs are within the range of the 95th percentile confidence interval of the univariate GEV distribution fit (figs. A2 and A3). Uncertainties associated with the copula analysis and following the "Maximum Density" approach do not extend too much from the median values and stay within the confidence interval of the univariate GEV distribution (figs. A2 and A3) for most of the copulas tested. Under this "Maximum Density" approach and based on the best copula fits, they differ by about 3-8 $m^3$ $s^{-1}$ for the 5-year RP and 2-9 $m^3$ $s^{-1}$ for the 30-year RP. Under those same conditions, the uncertainties from the "Weighted 425 Average" approach vary between 65 and 149 $m^3$ $s^{-1}$ for the 5-year RP and between 37 and 68 $m^3$ $s^{-1}$ for the 30-year RP. Therefore, uncertainties related to the "Maximum Density" approach associated with the most likely scenarios are relatively small, providing reasonable confidence in such scenarios. Conversely, the "Weighted Average" approach uncertainties provide a confidence interval on possibly more extreme scenarios, which is relevant when communicating RLs.

- ***Input datasets selection***

Depending on the choice of river time series as initial input, the results of the copula analysis under the 5- and 30-year RPs differ substantially around a maximum of 20 $m^3$ $s^{-1}$ and 40 $m^3$ $s^{-1}$, respectively, with an NDV range of 6-10% and 8.4-15% (fig. 8). This contrasts with the choice of sea level time series as initial input with a maximum difference of around 6 and 12 $m^3$ $s^{-1}$ equivalent to 1.5-3% and 2.3-4.6% NDV bounds for the 5- and 30-year RPs, respectively, without considering the three sets 435 associated with obs sea level. Those results are based on the most likely scenarios from each best copula fit and did not consider the obs sea level associated sets. It emphasizes that the choice of sea level records has a lower influence than the one of river discharge within this study on compound hydrological extreme events on our example study site (Halmstad). The well-recognized issues from the reanalysis dataset support this result as even a large difference in the sea level input dataset does not get reflected in the NDV values when looking at the choice of sea level. Similar findings could be expected for the 440 surrounding area (West coast of Sweden).

- ***Copula selection***

To evaluate the role played by choice of the copula, we calculated the NDV for each set between the maximum and the minimum values returned by the 18 copulas tested without considering the sets with obs sea level data input as it was too short 445 for bivariate analysis. Among all the different sets, the BB1, the Gaussian and the Clayton copulas are the best ones based on the different statistical criteria (section 2.2). Moreover, when only looking at the sets associated with the same river runoff input, the best copula fit is the same: Clayton for obs Nissan, BB1 for E-Hype except for S-Hype, which has Gaussian as the best fit for S-Hype/rec Halmstad and Clayton for the two other sets. The tests of using multiple copulas have also been investigated in previous studies. Lucey and Gallien (2022) looked at compound coastal events linking precipitation and/or sea

level in a tidal and semi-arid area. They noticed that, in their particular area, the Nelsen, BB1, BB5, and Roch-Alegre copulas represented best their datasets and each of them provided similar results in almost all cases. Bai et al. (2020) introduced a mixed copula which is a linear combination of Gumbel, Clayton, and Frank copulas to statistically study coastal winds and waves. They observed that the mixed copula can better describe the dependency structure than the five single copulas tested (Gaussian, t, Gumbel, Clayton, Frank) where the representation of relations between both drivers is complex.

For most of the sets, the "Fischer-Kock" and the "FGM" copulas give the highest RLs and the "Tawn" and "Joe" copulas give the smallest ones (figs. A2 and A3). It results in NDVs between 5.5% and 13% for the 5-year RP and between 3.8% and 9.5% for the 30-year RP. The base case E-Hype / rec Halmstad presents the highest NDVs when comparing with other sets NDVs, which emphasizes that the choice of copula is relatively more important than in other sets (fig. 8). Based on our assumption that this is possibly the best set, in terms of data sources, it stresses the idea that the choice of copula becomes more and more 460 critical when input datasets are long enough and statistically significant.

Therefore, the choice of copula has a similar influence as the choice of river discharge records for each of the nine sets tested here, as the obs sea level has not been considered. For both the 5- and 30-year RPs, the choice of sea level is the least impactful. Under the 5-year RP, the choice of copula is overall the most important before the choice of river discharge, but under the 30-year RP, the choice of river discharge predominates. However, this differs when looking at specific sets' copula NDVs (fig. 465 8).

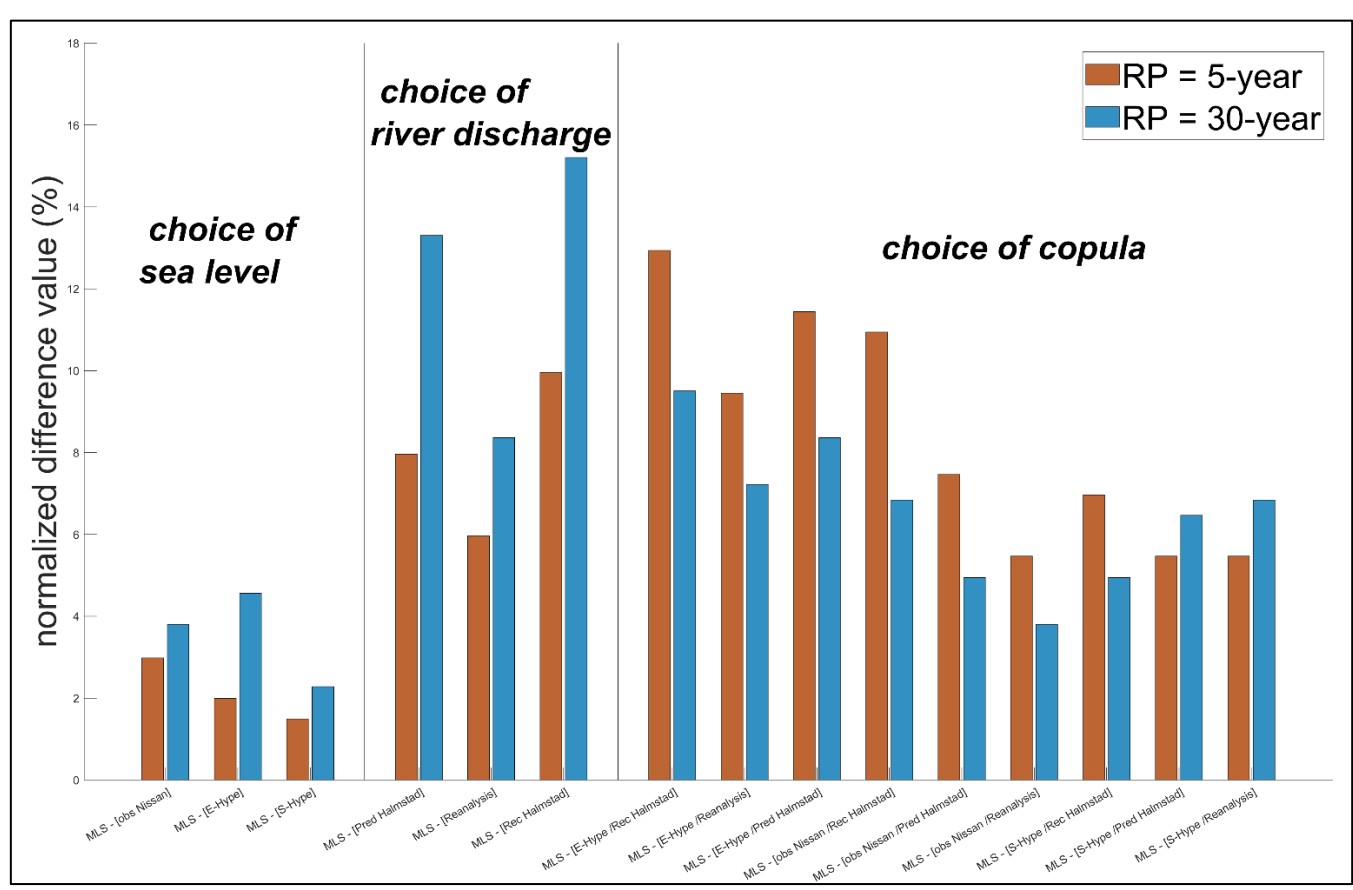

**Figure 8: Normalized difference values (%) for evaluating the importance of copula fit and forcing data for both 5- and 30-year return periods as mentioned in section 3.3.2.**

## 4 Limitations

Observed time-series datasets have a relatively short length, leading to rather high uncertainties once applying the GEV analysis. Similarly, model time-series datasets have inherent uncertainties, which can be challenging to quantify. Various data sources were assessed for their applicability in bivariate analysis, and direct sea level observations available for only 13 years were a limiting factor. We focus on longer reconstructed time series and other data sources for the principal analysis to explore uncertainties which are linked with available datasets of different lengths and biases. For example, re-analysis driven storm surge as well as different modelling approaches such as S-Hype and E-Hype models present some uncertainties due to the modelling nature of such datasets especially towards the extremes where they are often underestimated. Moreover, assumed stationarity within the datasets can be a limitation while performing the statistical analysis (Kudryavtseva et al., 2021) even though for the neighbour station of Ringhals it has been shown that non-stationary models were not statistically significant (Rydén, 2024). The choice of the sampling datasets based on annual maxima can be a limitation. For instance, in their specific

tidal dominated and semi-arid area, Lucey and Gallien (2022) stated that annual maximum sampling seems to underestimate water levels at longer RPs. In this study, only the compounding between sea level and river discharge has been studied but Latif and Simonovic (2023) showed that considering the three drivers storm surge, precipitation and river discharge to study compound coastal floods can provide a better statistical approach and therefore better estimate joint RPs in their study area located on the West Coast Canada. However, after carrying out a brief sensitivity analysis on defining extreme sea level events as sea level peaks above the 95[th] or 99[th] percentile and comparing it with the annual maxima sampling, no noticeable changes were found; a similar conclusion was also drawn by Ward et al. (2018).

A compound analysis is seen as a relatively new approach within this field of study, which also involves some limitations, such as the quantification of uncertainties within a multivariate analysis that differ widely depending on the choice of a copula. The uncertainty resulting from the choice of copula can to some extent be constrained by adopting appropriate goodness-of-fit statistics for the selection of the best-fitting copula. In this study, we choose to illustrate this indirectly by presenting results from many different choices of copula, despite having calculated such goodness-of-fit metrics (section 2.2). Furthermore, we showed the normalized difference values for different data sources. As discussed in section 2.3 and Serinaldi (2015) a careful interpretation comparing return levels from different hazard scenarios is, however, always needed. In decision-making adapting a strategy such as ours (to include results from all studied copulas and also different data sources) has some limitations in the sense that too much information can sometimes cause more confusion than help for the decision. Often it may be possible to argue against some choices of copulas (e.g. the Gaussian copula when the distributions are skewed) and the strategy of constraining the results to one copula or a set of "best-fitting" copulas using some threshold on the goodness-of-fit metrics may be appropriate. For the purpose of our study and the conclusion drawn we consider, however, that presenting the results from multiple choices of data and multiple copula is appropriate.

## 5 Summary and conclusions

This study assesses the hydrological and oceanographic processes that may lead to compound flood effects in Halmstad. The method is easily transferable to other regions or sites. In the paper, we stress the importance of the choice of data sources and copulas for multivariate analysis. Based on our analysis, we conclude that:

•	A dependency is found between the annual maxima of river discharge and the corresponding sea level. The dependency for annual sea level maxima and associated river discharge was not considered significant at this site.

•	All values of the "most likely" scenarios and their uncertainties resulting from the copula analysis are within the range of the 95[th] percentile confidence interval of the univariate GEV distribution fit.

•	The choice of river time series as initial input influences the results of the copula analysis to a higher degree than the choice of sea level time series as initial input.

•    Copula choice has a similar influence on return period statistics as the river discharge input for most of the twelve sets tried.

•    According to statistical criteria, the Clayton, BB1 and Gaussian (once) copulas performed the best in this study.

Uncertainties in compound flood hazard quantification are essential to consider. They can come from different sources, such as methodology and data sources. Each type of uncertainty from the individual components due to the length of the time series and the modelling ones is also propagated in multivariate risk estimation. This study highlighted the need to be careful when choosing such or such data sources in that regards as it may result in quite different outputs if only looking at one data source,

which inherently is associated with some uncertainties. Therefore, this study stresses the importance of the choice of data sources and copula.

## 5 Appendices

**Table A1: Rank correlation (rho) and p-values of the twelve different sets based on ($Q_n/s_n$) in columns: " / river" and ($q_n/S_n$) in columns " / sea" ; the best set of study is displayed with bold and underlined; p_values above 5% are highlighted in italic.**

| river | sea level | rank correlation | rho / river | p / river | rho / sea | p / sea |
|---|---|---|---|---|---|---|
| E-Hype | reanalysis | Pearson | 0.4532 | 0.0154 | *0.0443* | *0.8227* |
| E-Hype | reanalysis | Kendall | 0.3280 | 0.0141 | *0.0370* | *0.7992* |
| E-Hype | reanalysis | Spearman | 0.4532 | 0.0163 | *0.0443* | *0.8226* |
| obs Nissan | reanalysis | Pearson | *0.2430* | *0.2478* | *0.6684* | *0.0922* |
| obs Nissan | reanalysis | Kendall | *0.2896* | *0.1594* | *0.6410* | *0.0725* |
| obs Nissan | reanalysis | Spearman | *0.2419* | *0.2478* | *0.6674* | *0.0922* |
| S-Hype | reanalysis | Pearson | *0.3799* | *0.1325* | *-0.0760* | *0.7719* |
| S-Hype | reanalysis | Kendall | *0.3088* | *0.0914* | *-0.0588* | *0.7765* |
| S-Hype | reanalysis | Spearman | *0.3799* | *0.1333* | *-0.0760* | *0.7729* |
| E-Hype | observed | Pearson | *0.4725* | *0.1030* | *0.4056* | *0.1908* |
| E-Hype | observed | Kendall | *0.3590* | *0.1000* | *0.2424* | *0.3108* |
| E-Hype | observed | Spearman | *0.4725* | *0.1057* | *0.4056* | *0.1926* |
| obs Nissan | observed | Pearson | *0.4396* | *0.1329* | *-0.1049* | *0.7456* |
| obs Nissan | observed | Kendall | *0.3333* | *0.1289* | *-0.0606* | *0.8406* |
| obs Nissan | observed | Spearman | *0.4396* | *0.1350* | *-0.1049* | *0.7495* |
| S-Hype | observed | Pearson | 0.6273 | 0.0388 | *-0.0490* | *0.8799* |
| S-Hype | observed | Kendall | 0.4909 | 0.0405 | *-0.0303* | *0.9466* |
| S-Hype | observed | Spearman | 0.6273 | 0.0440 | *-0.0490* | *0.8863* |
| E-Hype | | Pearson | 0.4439 | 0.0109 | *0.2438* | *0.1788* |

| | | | | | | |
|---|---|---|---|---|---|---|
| | | Kendall | 0.3145 | 0.0111 | *0.1452* | *0.2518* |
| | | Spearman | 0.4439 | 0.0116 | *0.2438* | *0.1782* |
| obs Nissan | pred Viken | Pearson | *0.3400* | *0.1040* | 0.2435 | 0.2516 |
| | | Kendall | *0.2319* | *0.1189* | 0.1522 | 0.3128 |
| | | Spearman | *0.3400* | *0.1044* | 0.2435 | 0.2505 |
| S-Hype | | Pearson | *0.3676* | *0.1466* | 0.2843 | 0.2687 |
| | | Kendall | *0.2794* | *0.1288* | 0.1912 | 0.3081 |
| | | Spearman | *0.3676* | *0.1471* | 0.2843 | 0.2678 |
| *E-Hype* | *rec Viken* | Pearson | **0.4836** | **0.0044** | 0.2753 | 0.1273 |
| | | Kendall | **0.3523** | **0.0036** | 0.1815 | 0.1500 |
| | | Spearman | **0.4836** | **0.0048** | 0.2753 | 0.1272 |
| obs Nissan | | Pearson | *0.3446* | *0.0916* | 0.1487 | 0.4880 |
| | | Kendall | *0.2333* | *0.1076* | 0.1014 | 0.5071 |
| | | Spearman | *0.3446* | *0.0921* | 0.1487 | 0.4863 |
| S-Hype | | Pearson | *0.4882* | *0.0550* | 0.1373 | 0.5994 |
| | | Kendall | 0.3833 | 0.0413 | *0.1029* | *0.5976* |
| | | Spearman | *0.4882* | *0.0572* | 0.1373 | 0.5986 |

**Table A2: Summary report of runs from the copula analysis for the twelve different sets; the best study set is highlighted in bold, italic and underlined.**

| river | sea level | Copula best fit | number of co-occurring years |
|---|---|---|---|
| obs Nissan | reanalysis | Clayton | 24 |
| | obs sea level | Galambos | 13 |
| | pred Halmstad | Clayton | 24 |
| | rec Halmstad | Clayton | 25 |
| *E-Hype* | reanalysis | BB1 | 28 |
| | obs sea level | Gaussian | 13 |
| | pred Halmstad | BB1 | 32 |
| | *rec Halmstad* | *BB1* | *33* |
| S-Hype | reanalysis | Clayton | 17 |
| | obs sea level | BB1 | 11 |
| | pred Halmstad | Clayton | 17 |
| | rec Halmstad | Gaussian | 16 |

**Table A3: Summary report from the river discharge's results and associated uncertainties from the copula analysis for the E-Hype / rec Halmstad set; the results from the univariate method are highlighted in bold, italic and underlined. The Gaussian copula has not been considered for the analysis of the "most likely scenarios" row.**

| E-Hype / rec Halmstad | copula distribution fit | | 5-years RL | 30-years RL |
|---|---|---|---|---|
| *univariate* | | *median* | *201* | *263* |
| | | *max 95%* | *250* | *431* |
| | | *min 5%* | *167* | *189* |
| | | *relative error* | *41.29%* | *92.02%* |
| "most likely scenarios" | Fischer-Kock / FGM | max copula | 234 | 292 |
| | Tawn | min copula | 208 | 267 |
| | *NDV* | | *12.94%* | *9.51%* |
| uncertainties "Maximum Density" approach | BB1 | max without ouliers | 224 | 284 |
| | BB1 | min without ouliers | 218 | 276 |
| | BB1 | *NDV* | *2.99%* | *3.04%* |
| | Joe | max without ouliers | 217 | 274 |
| | Joe | min without ouliers | 210 | 268 |
| | Joe | *NDV* | *3.48%* | *2.28%* |
| | Fischer-Kosck | max without ouliers | 235 | 296 |
| | Fischer-Kosck | min without ouliers | 215 | 272 |
| | Fischer-Kosck | *NDV* | *9.95%* | *9.13%* |
| | Gaussian | max without ouliers | 220 | 398 |
| | Gaussian | min without ouliers | 218 | 262 |
| | Gaussian | *NDV* | *1.00%* | *51.71%* |
| uncertainties "Weighted Average" approach | BB1 | max without ouliers | 351 | 329 |
| | BB1 | min without ouliers | 202 | 261 |
| | BB1 | *NDV* | *74.13%* | *25.86%* |
| | Joe | max without ouliers | 328 | 303 |
| | Joe | min without ouliers | 202 | 261 |

| | Joe | NDV | 62.69% | 15.97% |
|---|---|---|---|---|
| | Fischer-Kosck | max without ouliers | 353 | 353 |
| | Fischer-Kosck | min without ouliers | 202 | 261 |
| | Fischer-Kosck | NDV | 75.12% | 34.98% |
| | Gaussian | max without ouliers | 349 | 338 |
| | Gaussian | min without ouliers | 202 | 261 |
| | Gaussian | NDV | 73.13% | 29.28% |

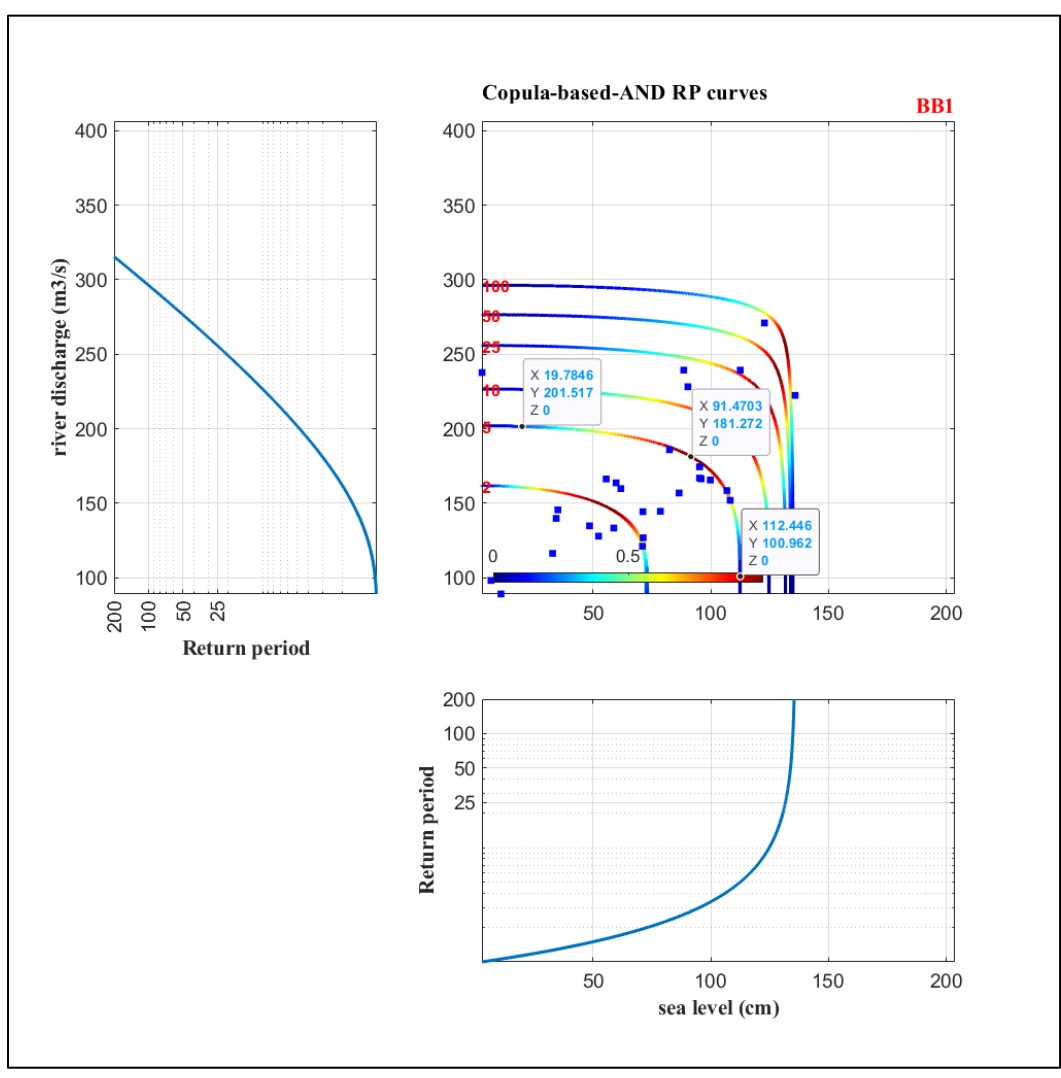

**Figure A1: Set E-Hype / rec Halmstad, best copula fit: BB1. [2, 5, 10, 25, 50, 100] RPs and associated densities. The left and lower panels correspond to marginal RPs curves of each univariate parameter individually, river discharge and sea level (extracted from MhAST software).**

**Fig. A2)**

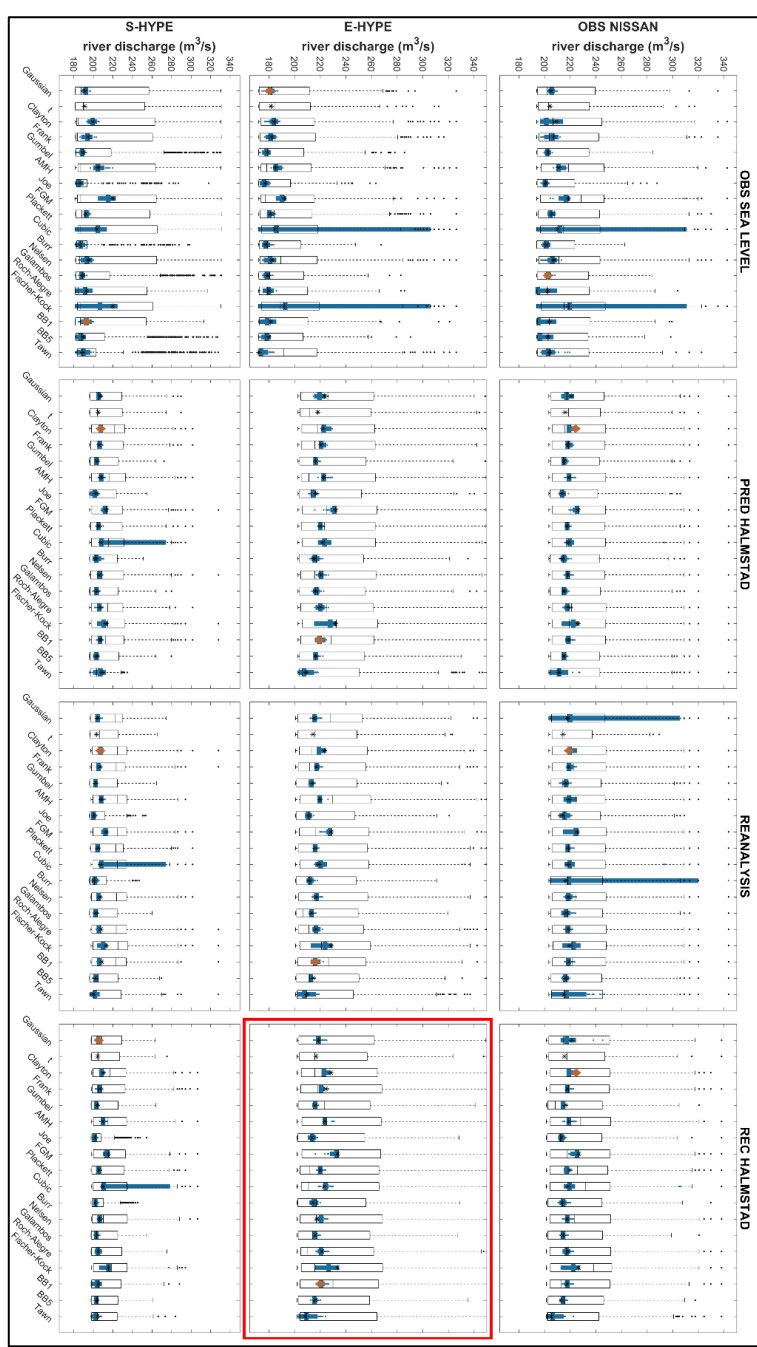

Figure A2: Fluvial component in bivariate events with 5-year RP values from copula fits. Each subplot corresponds to a set of events from an association of river discharge and sea level inputs displayed as a matrix, and in each column, a copula distribution fit where two uncertainty approaches are displayed as error bars. Stars represent the most likely scenarios return values from each copula for each set and each red diamond, the best-fit copula. The two uncertainty approaches are displayed as boxplots that give a statistical summary. Median, First and third quartiles are represented in each box; Whiskers represent minimum and maximum values, and dots represent outliers. Outlined boxplots correspond to the "Weighted Average" approach, and filled ones to the "Maximum Density" approach. The set E-Hype / Rec Halmstad, used as a base case, is highlighted by the red rectangles.

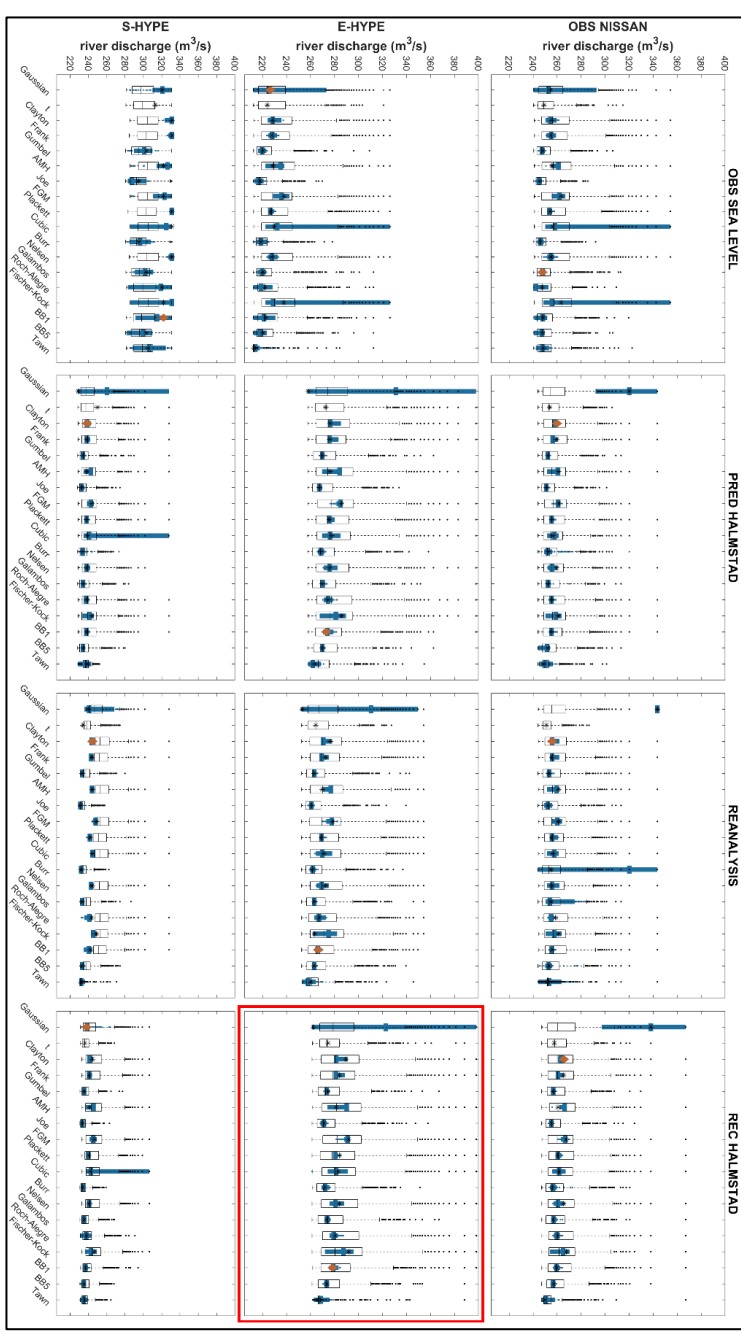

**Figure A3: Fluvial component in bivariate events with 30-year RP values from copula fits.** Each subplot corresponds to a set of events from an association of river discharge and sea level inputs displayed as a matrix, and in each column, a copula distribution fit where two uncertainty approaches are displayed as error bars. Stars represent the most likely scenarios return values from each copula for each set and each red diamond, the best-fit copula. The two uncertainty approaches are displayed as boxplots that give a statistical summary. Median, First and third quartiles are represented in each box; Whiskers represent minimum and maximum values, and dots represent outliers. Outlined boxplots correspond to the "Weighted Average" approach, and filled ones to the "Maximum Density" approach. The set E-Hype / Rec Halmstad, used as a base case, is highlighted by the red rectangles.

*Code availability.* The code used to generate the figures can be acquired by contacting the first author (kevin.dubois@geo.uu.se).

*Data availability.* Observations data are available from SMHI at https://www.smhi.se/data/oceanografi/ladda-ner-oceanografiska-observationer#param=sealevelrh2000,stations=core for the sea level (accessed on: 14th of October 2021) and https://vattenweb.smhi.se/station/ for the river discharge (accessed on: 28th of April 2021).

*Author contributions.* KD conducted the analysis. KD prepared the manuscript with contributions from all co-authors.

*Competing interests.* The authors declare that they have no conflict of interests.

*Acknowledgements.* The work forms part of the project: Extreme events in the coastal zone – a multidisciplinary approach for better preparedness. The authors would like to thank the three anonymous reviewers for their valuable comments which contributed to improving the quality of the paper. We further thank Johanna Mård, Christoffer Hallgren and Faranak Tootoonchi for our useful discussions.

*Financial support.* This research was funded by the Swedish Research Council FORMAS (Grant No. 2018-01784) and the Centre of Natural Hazards and Disaster Science (CNDS).

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
