# Peer review of "Influence of data source and copula statistics on estimates of compound flood extremes in a river mouth environment"

_Natural Hazards and Earth System Sciences, 2023_

## Author Comment (AC1)

I only really have two comments, but they are potentially pretty major ones:

Section 2.1.1 – I'm not convinced by the accuracy of the sea level data set, since the observed and reanalysis extremes do not correspond well with each other and then the reconstructed data is trained on these contrasting datasets. There are many sites globally where a longer accurate observational sea level record exists, so why not chose a different site?

The idea proposed here is to get as local as possible and extract data from a longer available time series from another station would lead us to not so local sea level conditions. That is why we carried out a reconstructed time series. The reconstructed data is only based on observations, the reanalysis data has known issues that makes it difficult to work with. However, when it comes to carrying this sensitivity analysis, we consider it is not a major issue as the sea level data does not seem to have a strong impact on the copula results and even more strengthens our conclusion that, for this case, input hydrological data influences the results the most. Here, Halmstad has been chosen as the site is potentially prone to compound events on the Swedish coast where the highest sea level has been recorded so far (lines. 65 to 68). In addition, the Swedish west coast has in previous work been found to be an important area to study (Andersson, 2021; Hieronymus and Kalen, 2020) due to its multiple aspects causing risk for flooding. To help guiding and communicating with the local municipalities about their continued work to protect coastal areas from flooding we considered it useful to pick one site in this area as an example to showcase the applied methods and their results. Halmstad was then further decided upon because it has the highest observed sea level of all the Swedish measurement stations. We do not expect significantly different results from other sites in this area. We understand the reviewer's comment that other sites globally on completely different geographic areas could also be studied, but this would not necessarily be informative for the local focus area and we will clarify this aspect and motivate our choice of study area more clearly in the revised manuscript (lines 62 to 73).

In the end, you produce a 44 year record of sea level variability. Despite this being a long data set, you only select the annual extremes for the analysis - why do this when in effect this reduces this large data set down to only 44 (suspect) data points. Since the purpose is to assess joint probabilities, this could be done on a larger subset of extremes, e.g. >99$^{th}$ percentile peaks.

We decided to only use the annual extremes for the analysis which is a common approach in the literature. However, we also carried out a really brief analysis on defining extreme events as sea level above the 95$^{th}$ percentile value and another test in using the threshold value of the 99$^{th}$ percentile. This brief analysis did not seem to make any difference in our conclusions as also found in Ward et al., 2018; but a more extended sensitivity analysis could be, we think, highly relevant. However, we believe this is outside the scope of this study. We also refer to this point in the section 4. Limitations.

Ward, P. J., Couasnon, A., Eilander, D., Haigh, I. D., Hendry, A., Muis, S., Veldkamp, T. I. E., Winsemius, H. C., and Wahl, T. (2018) Dependence between high sea-level and high river discharge increases flood hazard in global deltas and estuaries, Environmental Research Letters, 13(8), 084012. 10.1088/1748-

9326/aad400.Andersson, M.: Climate Adaptation by Managed Realignment. Future mean and extreme sea levels, SMHI, Report number: 2021/912/9.5, 16–17, 2021.

Hieronymus, M. and Kalén, O.: Sea-level rise projections for Sweden based on the new IPCC special report: The ocean and cryosphere in a changing climate, Ambio, 49, 1587–1600, https://doi.org/10.1007/s13280-019-01313-8, 2020.

---

## Author Comment (AC2)

The manuscript proposes a flexible framework for the attribution of the uncertainties associated with joint exceedance probability estimates of river discharge -coastal water levels. The framework is demonstrated at a case study site on the west coast of Sweden. Copula family and the dataset chosen to represent river discharge are found to exert the largest influence on the estimates. The manuscript is overall well written, topical, and the results are interesting, however, I do have several reservations about accepting in its present form. Key literature is missing, the discussion section is subpar, and the novelty of the study is debatable.

Thank you for your review and many helpful comments to improve our study. According to our knowledge, the novelty of this study comes from the sensitivity analysis focusing on the influence of using different data sources in this particular context of compound coastal flood. Our conclusion highlights the need for communicating uncertainties depending on datasets used in such analysis. For example, in places where local data are unavailable, the use of available global data as input can result in large results biases which therefore can lead to rather important impacts for coastal applications and management. We hope our responses given below marked in red as well as our changes in the manuscript have helped to address these issues.

Updated references for section 2.1.1 have been made due to final adjustments to the now published methodology presented in Dubois et al. (2024).

**General comments**

Title is misleading since no river mouth water levels are calculated.

We agree and have adjusted the manuscript by changing the title as follows: "Influence of data source and copula statistics on estimates of compound flood extremes in a river mouth environment".

The first paragraph, although not incorrect, is odd in the sense that it stresses that heavy precipitation, storm surge and runoff can be caused by different weather conditions when a key rational for the statistical dependence is that the flood drivers are forced by the same large scale weather conditions.

We think, your comment relates to our miscommunication with this first paragraph as our communication aimed to introduce different processes that could result in floods while keeping in mind that we are further interested in compound events forced by the same large-scale weather conditions as you mentioned. We therefore adjusted it to clarify it (lines 25-30).

A more detailed description of the "Weighted Average" and a "Maximum Density" approach in the MhAST toolbox is required for readers unfamiliar with the toolbox.

The paragraph mentioning the description of those 2 approaches (Lines 223-231) has been extended for readers unfamiliar with the toolbox.

I do not understand why there is an entire section on univariate (oceanic and fluvial) flooding when the investigation is about compound events. The return levels in the boxplots (Figure 6) are not estimates of the 5- and 30- year fluvial events, they are the fluvial component in bivariate events with those return periods. I am unsure as to whether the bivariate and univariate return periods should be compared and whether

statements such as "Moreover, the RL uncertainties for the "Maximum Density" approach are all located within the 95th confidence interval of the univariate RL." are meaningful.

As the common practice for stakeholders is based on univariate flooding, we believe it is important to implement it within this type of study. Moreover, we think it brings a certain understanding of the datasets and increases the readability of the paper when it comes to the framing of this study, especially when it later comes to comparing the different data source influences. We agreed and precise our caption (figure 6 and figure A.1) which was missing clarity.

We do not think the bivariate and univariate should, in this study, be directly compared (lines 251-253). However, we think both approaches provide important information especially when investigating uncertainty as presenting results from both can bring a more comprehensive idea of the role of uncertainties in that particular context.

The discussion should compare the findings with other similar studies, see Lucey and Gallien (2022) and Santos et al. (2021) for starters.

We agree with you that comparing our findings with other studies is relevant and we therefore followed your suggestion so we added sentences referring to previous work to stimulate the discussion (lines 279-280 -> Bevacqua et al. (2017) / lines 392-393 -> Santos et al. (2021) / 423-425 & 451-453 -> Lucey and Gallien (2022) / 426-428 -> Bai et al. (2020) / 453-456 -> Latif and Simonovic (2023)).

**Specific comments**

L13: Statistical copulas do not give a measure of flood risk (at least not directly).

We agree and have adjusted the manuscript by adding the word "indirectly".

L35: There are a great many other studies that examine the dependence between river discharge and storm surge at sites in Europe that should be cited here (e.g., Hendry et al. 2019, Ward et al. 2018).

Thank you for your comment, we have adjusted the manuscript accordingly (lines 41-46).

L35, L71 and elsewhere: Be careful to specify that these "interactions" refer to their co-occurrence probabilities and not physical interactions. This would be a good place to introduce frameworks that link statistical and numerical models to account for joint exceedance probabilities and physical interaction to locate the stretches of river where compound flooding is an issue (e.g., Moftakhari et al. 2019, Gori et al. 2020, Jane et al. 2022). Studies such as Couasnon et al. (2020) and Moftakhari et al. (2017) only carry out statistical modeling and therefore only assess the "potential for compound flooding", they do not determine "impacts from compound flooding" either in terms of estimating water level or computing inundation depths.

We agree and specified that these interactions refer to their co-occurrence probabilities, at least when we introduce this word the first time. We focused on the already introduced references as we only looked into the statistical approach and therefore the "potential for compound flooding" but we agree that such literature is relevant to add and can improve

the clarity to this study. We then have adjusted the manuscript to clarify this point (lines 87-92).

L46: Reference required.

This paragraph was focused on the study from Bevacqua et al. (2019). I then added the reference there too and added some precision to the physical processes studied.

L73: Sentence implies there is a single annual maximum value for each year but of course each year will possess a different annual maximum value.

Thank you, you are correct. We added a sentence to clarify and report the range of variation in annual maximum values.

L76: Technically since the distributions are continuous this would be an "exceedance probability".

Thank you for your comment, we then adjusted the manuscript accordingly.

L79: "variable" or "driver" is potentially more accurate language than "factor".

We agree and have adjusted the manuscript changing "variable" by "driver".

L81: Is "potential for compound events" more accurate than "a potential compound event".

We agree this is more accurate and have adjusted the manuscript.

L98: What was the other dataset used in the correlation analysis? Also change "statistically insignificant" to "not statistically significant".

We agree and have adjusted the manuscript. The other dataset used in the correlation analysis is primarily the observations river discharge and then the E-Hype and S-Hype models. This is introduced later in the paper.

L167 & 169: Could say "paired" instead of "used" as the latter is "used" a lot throughout the paper!

We agree and have adjusted the manuscript.

L170: Move references to the end of the sentence.

We agree and have adjusted the manuscript.

L185: Null hypothesis in such tests is usually that the correlation coefficients are zero indicating it is reasonable to assume the variables are independent.

Indeed but we still think it is interesting to briefly check the Null hypothesis when possible.

L213: "Adopting the "AND scenario" (see above) permitted us to investigate the dependency between sea level and river discharge during extreme events." The "OR" HS also allow this!

We agreed and therefore clarified this point that we have been interested in looking into the risk of compound events only (AND scenario) and used this opportunity to look into the dependency between both variables. We rewrote this sentence as follows "Adopting the "AND scenario" (see above) permitted us to investigate the risk of compound events only highlighting the dependency between sea level and river discharge during extreme events.".

L237: "In the following, we mainly focus on the "OR scenario" yet in the next paragraph, only  the "AND scenario" is discussed! Justification for the "OR scenario" is poor here. By "compound flood risk driven regardless of the situation (oceanographic or hydrological)" I believe you mean you're interested in compound risk and risk from the oceanographic only and hydrological only events.

We agreed and rewrote to clarify it as follows "In the following subsection 3.1, we look into the "AND scenario" as we investigate the compound risk only. In the subsections 3.2 and 3.3, we mainly focus on the "OR scenario"  (see above) as we … ".

L245, 337: The term "superposed" implies a decision taken by the practitioner "do not overlap" maybe a clearer description.

We agree and have adjusted the manuscript.

L272: "The BB1 copula fit has a 5-year RL of 220 m3/s." is not correct the BB1 copula fit will have many discharge values associated with a 5-year RL, that depend on the corresponding sea level.

We agreed and rewrote as follows "The BB1 copula fit has a 5-year RL "most likely scenario" of 220m$^3$/s".

L272: What copulas are you referring to here?

We are here referring to all the copulas fits we have tested for. We slightly rewrote the sentence to clarify this point as follows "Among all tested copulas, their 5-years RLs …".

L296: Please explain what the higher return levels are being compared with.

We agreed this sentence is somewhat vague and lacks clarification, we decided to delete "with higher RLs when considering the compound effects" from it as our goal is to highlight that the copulas and uncertainties present similar behaviour.

L335: "more significant effect on estimated RLs" more significant effect than what?

This refers to switching data sources that may have a more significant effect on estimated RLs than switching method approaches but we agreed it can lead to confusion and we deleted the word "more" to avoid such confusion.

L339: Consider re-writing: "This similarity stresses the idea that river discharge predominates over sea-level inputs." Since the phrase "stresses the idea" is sort of ambiguous furthermore I wonder whether "dominates" is more suitable than "predominates".

We agreed and rewritten this sentence as follows "This similarity emphasizes that river discharge dominates over sea-level inputs".

L349: "This study focuses on extreme hydrological events associated with oceanographic conditions and, therefore, concentrates on the RLs of river discharge." I do not understand the point trying to be made here!

This point might create confusion but aims to direct the reader towards the fact we are looking into RLs of river discharge "only" even though we are studying compound events and we could look into its associated sea level component as well but because we only noticed a significant correlation for annual river discharge and associated sea level (table A1),.we consider the river discharge as the main variable to focus on.

L371: "results" is ambiguous. Is this the "most likely" event?

We are here relating the word "results" to the fact that the choice of sea level records has a lower influence than the one of river discharge. We therefore decided to change the word "results" by the word "findings".

L417: "The opposite dependency" is a strange turn of phrase.

We agree and deleted the word "opposite" which might create confusion.

L424: Consider changing "The choice of copula has a similar magnitude of its influence on return period statistics as the choice of river discharge input for most of the twelve sets tried" to "Copula choice has a similar influence on return period statistics as the river discharge input for most of the twelve sets tried".

We agree and have adjusted the manuscript accordingly.

Appendices: Change "," to "." (decimal places).

We agree and have adjusted the manuscript.

**References**

Gori, A., Lin, N., and Xi, D. (2020). Tropical cyclone compound flood hazard assessment: From investigating drivers to quantifying extreme water levels. Earth's Future, 8, e2020EF001660. https://doi.org/10.1029/2020EF001660.

Hendry, A., Haigh, I. D., Nicholls, R. J., Winter, H., Neal, R., Wahl, T., Joly-Laugel, A., and Darby, S. E. (2019) Assessing the characteristics and drivers of compound flooding events around the UK coast, Hydrology and Earth System Science, 23, 3117–3139. https://doi.org/10.5194/hess-23-3117-2019.

Jane, R., Santos, V. M., Rashid, M. M., Doebele, L., Wahl, T., Timmers, S. R., Serafin, K. A., Schmied, L., and Lindemer, C. (2022) A Hybrid Framework for Rapidly Locating Transition Zones: a Comparison of Event- and Response-based Return Water Levels in the Suwannee River FL, Water Resources Research, 58, e2022WR032481. https://doi.org/10.1029/2022WR032481.

Lucey, J. T. D. and Gallien, T. W. (2022) Characterizing multivariate coastal flooding events in a semi-arid region: the implications of copula choice, sampling, and infrastructure, Natatural Hazards and Earth System Science, 22, 2145–2167. https://doi.org/10.5194/nhess-22-2145-2022.

Moftakhari, H., Schubert, J. E., AghaKouchak A., Matthew, R. A., and Sanders, B. F. (2019) Linking statistical and hydrodynamic modeling for compound flood hazard assessment in tidal channels and estuaries, Advances in Water Resources, 128, 28-38. https://doi.org/10.1016/j.advwatres.2019.04.009.

Santos, V. M., Casas-Prat, M., Poschlod, B., Ragno, E., van den Hurk, B., Hao, Z., Kalmár, T., Zhu, L., and Najafi, H. (2021) Statistical modelling and climate variability of compound surge and precipitation events in a managed water system: a case study in the Netherlands, Hydrol. Earth Syst. Sci., 25, 3595–3615. https://doi.org/10.5194/hess-25-3595-2021.

Ward, P. J., Couasnon, A., Eilander, D., Haigh, I. D., Hendry, A., Muis, S., Veldkamp, T. I. E., Winsemius, H. C., and Wahl, T. (2018) Dependence between high sea-level and high river discharge increases flood hazard in global deltas and estuaries, Environmental Research Letters, 13(8), 084012. 10.1088/1748-9326/aad400.

Bai, X., Jiang, H., Li, C., and Huang, L.: Joint probability distribution of coastal winds and waves using a log-transformed kernel density estimation and mixed copula approach, Ocean Eng., 216, https://doi.org/10.1016/j.oceaneng.2020.107937, 2020.

Bevacqua, E., Maraun, D., Hobæk Haff, I., Widmann, M., and Vrac, M.: Multivariate statistical modelling of compound events via pair-copula constructions: Analysis of floods in Ravenna (Italy), Hydrol. Earth Syst. Sci., 21, 2701–2723, https://doi.org/10.5194/hess-21-2701-2017, 2017.

Latif, S. and Simonovic, S. P.: Compounding joint impact of rainfall, storm surge and river discharge on coastal flood risk: an approach based on 3D fully nested Archimedean copulas, Springer Berlin Heidelberg, 1–32 pp., https://doi.org/10.1007/s12665-022-10719-9, 2023.

Olbert, A. I., Moradian, S., Nash, S., Comer, J., Kazmierczak, B., Falconer, R. A., and Hartnett, M.: Combined statistical and hydrodynamic modelling of compound flooding in coastal areas - Methodology and application, J. Hydrol., 620, 129383, https://doi.org/10.1016/j.jhydrol.2023.129383, 2023.

---

## Author Comment (AC3)

The paper presents the compound flood risk analysis across the Swedish coast in the presence of low record availability and the choice of copula. While the uncertainty due to the first can't be averted, the second can be improved by the appropriate choice of copula and its parameter. Often sentences are not clear and require attention in framing. In a few cases, the methodology adopted is not robust and needs a relook. Often, there are misleading interpretations that make the paper weak. The paper can be published after appropriate revisions. The reviews are summarized as below:

Thank you for your review and many helpful comments to improve our study. We agree with your point that the uncertainty due to low record availability can't be averted directly, however this paper aims to argue that using different data sources can be highly important to better estimate uncertainties linked to the available datasets (as often the length of observations is short but modelled data have inherently uncertainties and biases). We hope our responses given below marked in red as well as our changes in the manuscript have helped to address these and other issues raised, also with regards to the interpretations and conclusions made.

Updated references for section 2.1.1 have been made due to final adjustments to the now published methodology presented in Dubois et al. (2024).

1. In Abstract, line 12: "The compound flood risks…. Often estimated using statistical copulas". This line can be misleading since copulas are one of the methods for estimating joint probability between two random variables. There are other methods as well, for example, joint entropy, or bivariate distributions considering box-cox transformations of associated random variables. Please consider revising/discarding this sentence.

Thank you for your comment, we have adjusted the manuscript accordingly (lines 12-13).

2. Line 27: What about the coastal backwater effects that influence the occurrence of compound flooding?

Thank you for your comment, this is actually what we have tried referring to when pointing out the relation to the storm surges effect and therefore clarified this point in adding a sentence about it and a reference (lines 30-32).

3. Please use the SI unit for sea level measurement.

Thank you for your comment, we do not see the issue of using cm as the sea level measurement unit as it is done in many other studies because the sea level variations are rather low (maximum around a few meters).

4. Line 82: Please use the word 'copula' throughout and not the 'statistical copula'.

Thank you for your comment, we agree and have adjusted the manuscript accordingly.

5. For processing 13-year sea level observation, a re-analysis coupled observational analysis was performed. In cases of data scarcity, the peak-over-threshold (POT) approach is in use instead of annual maxima. On the other hand, coupling different data sources, as adopted in this study, often results in

underestimation due to scale mismatch issues and extremes often underestimated in gridded reanalysis runs. If you are purely interested in observational assessment, the POT approach may be more powerful considering on average 2-3 to events per year, as compared to mixing reanalysis runs with the local tide gauge records.

As mentioned also to reviewer 1, we decided to only use the annual extremes for the analysis, which is a common approach in the literature, also to maintain the same method across the different datasets used for the paper for a better comparison basis. Concerning the sea level univariate brief analysis (as this is not the core of the paper but is rather used to introduce each dataset and assess differences independently between each of them), we did not mix any reanalysis data with tide gauge observations data. The reconstructed time series data is only based on observations (see reference to Dubois et al. (2024) that now got accepted for final publication). The reanalysis data has well-recognized issues that makes it difficult to work with at this level of detail. However, when it comes to carrying out the sensitivity analysis, we do not see this as a major issue as the sea level data do not seem to have a strong impact on the copula results and this issue even strengthens our conclusion that, for this case, hydrological data influence the results the most. We adjusted the manuscript to clarify this point (lines. 414-416).

However, for the copula analysis, we also carried out a brief analysis on defining extreme events at sea level above the 95th percentile value and another test using the threshold value of the 99th percentile. This analysis did not seem to make any difference in our conclusions as also found in Ward et al., 2018; but a more extended sensitivity analysis could be, we think, highly relevant. However, we believe this is outside the scope of this study. We also refer to this point in the section 4. Limitations.

On page 5, line 110-125: how you have converted hourly records to daily? The tide gauge records in Sweden are available at a minute-scale temporal resolution.

We converted downloaded hourly sea level tide gauge data from the SMHI download webpage to daily time series using the maximum hourly data within the day. We then adjusted accordingly the manuscript lines 115-116 & 126-127.

1. On Fig.3: lower panel, clearly shows that the reanalysis-driven reconstructed sea level observations are largely underestimated, especially at larger return period values. Please show the sea-level observation measurement in meters (SI unit).

We agree with you that, in terms of median values, the model-based and reanalysis datasets seem to largely underestimate return levels. However, it is important to keep in mind that the observations, as well, are associated with large uncertainties as displayed by the background colours and the background colours of the other sets include also the median RLs of the observations datasets (for more information, refer to Dubois et al., 2024). To keep consistency through the study, we rather would like to keep the sea level unit as cm.

2. between lines #145-150: What are the different sources of uncertainty of these models? Please describe number of parameters involve for calibration, forcing

data requirements and their temporal resolution. The predictive skills of the hydrologic models in simulating daily river discharge are not discussed at all.

Thank you for your comment, indeed we did not discuss in detail the hydrological models as that information can be found within the references that can be found within the manuscript. Some clarifying sentences have however been added to the manuscript to be clearer about where the interested reader can find evaluation and more details about the hydrological modelling (section 2.1.2).

3. Fig. 4: Y axis label: use superscript for the discharge measurement. Further, the uncertainty estimates between E-Hype and S-Hype model can be quantitatively estimated by the ratio of upper bound to the lower bound across higher and lower return levels.

Thank you for your suggestion, we adjusted the y-axis label. Indeed, quantitatively estimating the uncertainties estimates between E-Hype and S-Hype can be really interesting, but we believe that this is outside the scope of this specific study focusing on the sensitivity analysis of compound flood events analysed within a copula approach.

4. Line 177-178: Is it maximum likelihood based estimates of GEV parameters? This might be problematic for estimation of shape-parameters of GEV. Often a Bayesian estimate is proposed.

We indeed used the maximum likelihood to estimate GEV parameters (for the univariate study) and we agree a Bayesian estimate might be better. However, as also mentioned to reviewer 2, the univariate analysis is, we think, important in order to add clarity to the paper but not as a core part of the study. Therefore, we think it was not necessary to follow this analysis path and rather maintain it at the current level of simplicity.

5. Lines 203-206 and elsewhere: sentences are erroneous, please consider revising. Both 'OR' and 'AND' approaches are suitable for modelling joint effect: while the former consider a time offset, the later considers co-occurrence.

Both approaches used here correspond to the ones in Serinaldi (2015) where the "OR" approach accounts for both for a time offset where only one of the variables is high enough to create a bivariate occurrence hazardous but also accounts for a co-occurrence where both drivers are high enough to make a bivariate occurrence hazardous.

6. On page 11: line 245: highlights a 'discrepancy'.

As here, the dash lines assume independence between both drivers (annual river discharge and corresponding sea level) and the full yellow lines assume dependence between both drivers using the copula; then the fact that both lines are not superposed highlight that both variables are dependent otherwise they would be superposed. To clarify this point, the sentence has been revised (line 278).

7. Line 251: One do not assign any probability density function to each copula rather derives copula-based joint PDF.

Thank you for your comment, we agree with you and have adjusted the manuscript accordingly.

8. On page 13: line 280 onwards – this section and the subsequent ones are very confusing, rather much simpler and statistically robust methods should be adopted. The Gaussian copulas are not good while considering highly skewed data as here. The best method to select copulas are to apply the minimum AIC criteria with small sample corrections (in presence of limited data availability) followed by an appropriate goodness-of-fit measure, such as application of resample-based Cramer von Mises goodness-of-fit statistics.

This section is central to the paper and looks at the sensitivity of the choice of copula and aims to highlight the importance and challenges of choosing the best copula as you are suggesting and therefore, we would argue that including a large set of different copulas is important for proper context and discussion. Not because we argue that Gaussian fits are proper in this context. The method you proposed here, based on the AIC criteria, is indeed the one we use to select the best copula (see section 2.2 and corresponding references in this section) and we therefore agree with your comment. To clarify this point, precision has been added (line 300). While adopting different sampling strategies can be highly relevant, we decided to keep the same sampling strategy based on annual maxima across each dataset for consistency. We also conducted a brief analysis based on sampling values above the 95th and 99th percentiles but this did not seem to impact the analysis. However, a deeper sensitivity analysis on the sampling method could be really interesting and confirm this point but we think this is outside the scope of this study (see previous answer to comment 5).

9. Lines 335-340: Please explain in terms of hazards.

Thank you for your comment, we hopefully clarified this point (lines 358-359).

10. Line 339: Coincidence of independence line versus copula-derived dependence PDF does not necessarily stress the hypothesis that river discharge predominates over high sea levels. The other way around can also be possible.

Your comment seems to result from a misunderstanding of our paragraph as here, we refer to „corresponding dashed and full lines across the sets" and we do not state that independence line versus copula-derived dependence PDF coincide, rather the opposite actually (see previous paragraph). We regret this misunderstanding and have modified this section in the manuscript for this not to be carried forward to other potential readers. As we can see on Fig. 7, for example the sets E-Hype / rec Halmstad and the set E-Hype / pred Halmstad, their independence lines of both sets are almost superposing as well as their copula-derived dependence PDF. And this, as it is also the case for the datasets obs Nissan / pred Halmstad and obs Nissan / rec Halmstad stresses that river discharge dominates over sea level inputs.

11. Line 346: What is the 'most likely scenarios' here?

Each time we refer to the ‚most likely scenario', we refer to the definition given in the methodology section 2.2 (this has been elaborated in the manuscript in lines 227-232) that we extended slightly to clarify those points. So here, we refer to the

scenario from the best copula fit (according to statistical criteria as AIC) with the highest density along the closed-form joint probability density function of the copula.

12. Line 390 and associated section: There are several uncertainties in return levels due to the incorrect and erroneous application of copulas. Please use an appropriate goodness-of-fit measure to select the best-fit distribution. Also, there is not enough evidence that the SL is least sensitive to compound flood hazards; – mere little shift in density contours does not justify this major finding.

As mentioned previously, we fully agree with this general approach and therefore did use the AIC methodology, as well as other statistical tests, to rank copulas (section 2.2) but we decided to keep all of them to highlight the importance of the choice of copula. Here, in this particular case, the study did not find significant dependency between sea level annual maxima and corresponding river discharge and we only found a significant dependency between annual river discharge and corresponding sea level (section 3, first paragraph & table A.1). Figure A.2 resume all of our results and Fig. 8 shows the sensitivity test. This sensitivity test on switching data sources is, we think, a strong enough evidence that, in our case, the sea level is least sensitive to compound flood hazards as, for one set of river discharge fixed, the results across the different associated sea level datasets do not change drastically (low NDV) compared to fixing a set of sea level data. For our studied site and area, we consider that our result and conclusion are robust. On other sites and regions, the results could however be different and we encourage that more studies may be needed and we have carefully checked the revised manuscript to hopefully reflect in a reasonable way that uncertainty and limitations exist, as is unavoidable in most studies. Please see also revisions related to other reviewer comments.

13. In section 4: first paragraph, what is the need of extreme sea level analysis using model-derived sea level observations? A purely observational assessment employing different sampling mechanisms can work too. In the second paragraph, the uncertainty resulting from the choice of copula can be constrained by adopting appropriate goodness-of-fit statistics for the selection of the best-fitting copula.

Thank you for your comment, good point; we have added clarification to this section on the limitations of our study. The need for extreme sea level analysis using model-derived sea level observations in this study was motivated by the short available time series at the station of interest (13 years) which, in the Extreme Value Theory would be associated with really high uncertainties resulting in difficulties to draw any reliable conclusion towards longer return periods. Also, the goal of this paper is to highlight the risk of using only one type of data sources which has inherent limitations as well as one potentially wrong copula as it has been seen in previous literature. Your point on adopting the appropriate framework to select the best-fitting criteria is indeed in agreement with our conclusion where we highlight the importance of the choice of copula.

Ward, P. J., Couasnon, A., Eilander, D., Haigh, I. D., Hendry, A., Muis, S., Veldkamp, T. I. E., Winsemius, H. C., and Wahl, T. (2018) Dependence between high sea-level and high river discharge increases flood hazard in global deltas and estuaries, Environmental Research Letters, 13(8), 084012. 10.1088/1748-9326/aad400.

---

## Author Response (AR2)

The authors did a good job of addressing my concerns (Reviewer 2). I agree the novelty of this study stems from testing the sensitivity of the estimated hazard to the different data sources. I've listed a few minor comments below, in my view once these comments are addressed the manuscript will be acceptable for publication. Line numbers refer to the revised manuscript with track changes.

Thank you for your review and many helpful comments to improve our study. We hope our responses given below marked in red as well as our changes in the manuscript have helped to address these issues.

**Comments**

In the literature, "copula-based joint probability density function" are usually referred to as joint probability contours or isolines.

Thank you for your comment, we adjusted the manuscript accordingly.

Please clarify what the percentages stated in the abstract refer to.

Thank you for your comment, we adjusted the abstract in the manuscript accordingly (lines 20-21).

L128: Again, state the null hypothesis being tested as this will inform the reader of the correlation metric being used. The time series used to represent the river discharge also needs to be clarified.

Thank you for your comment, we adjusted the manuscript accordingly. As the correlation analysis has been explained in section 2.2 Bivariate analysis, we deleted this section to not confuse the reader.

L177: Perhaps the "model performs better .." is more appropriate than "performs best".

Thank you for your comment, we adjusted the manuscript accordingly.

L231: Thank you for adding more explanation on the terms "Weighted Average" and "Maximum Density" approach". It is however still unclear to me how the samples are generated for the two approaches. For the "Weighted Average" approach please expand "The weights are determined based on the critical joint RP." i.e., how are the weights calculated? Or is this sentence just stating how the joint probability contour is derived, if so, how do you obtain a sample from the contour. For the "most likely scenario", how is the density calculated? In other studies, such as Moftakhari et al. (2019) the probability density is estimated based on the observed sample. What do you mean by "adjusting these parameters" surely once the copula is fitted the parameters are fixed.

Thank you for your comment, we adjusted the manuscript accordingly, hopefully adding clarity to this section based on the method from Sadegh et al. (2018) in rewriting the text between lines 237 and 254 (track changes document).

L254: Add a comma after "only".

Thank you for your comment, we adjusted the manuscript accordingly.

L277: Could remove "the following".

Thank you for your comment, we adjusted the manuscript accordingly.

L279: Consider amending "in the compound flood risk driven regardless of the situation (oceanographic or hydrological)." to "in the total flood risk driven regardless of the situation (oceanographic, hydrological or compound)/"

Thank you for your comment, we adjusted the manuscript accordingly.

L283: Please provide justification for using the rec Halmstad / E-Hype set. I assume from Table A2 it is due to the length of the two time series.

Thank you for your comment, part of this choice is already introduced and motivated in the manuscript (lines 167-171 and lines 200-204) but we adjusted the manuscript to remind the reader about this choice (lines 296-297 in the track changes document).

L286 and in Figure 5 caption: Change "both" to "the".

Thank you for your comment, we adjusted the manuscript accordingly.

L381: "The differences between solid and dashed lines in Figure 7 are typically contained within about 20 cm sea level or 25 m3 s -1m3 /s river discharge, constituting about 10-15% of the extreme 5- and 30-year RLs for the site with a gap increasing with higher RPs." This presumably varies greatly along the contour and is therefore subjective depending on where on the contour you choose to look. Defining the maximum distance between the copula and independence cases on rays coming from the origin is potentially a more robust approach.

Thank you for your comment, we adjusted the manuscript accordingly.

L481: "As discussed in section 2.3 and Serinaldi (2015) a careful interpretation comparing return levels from different methodologies is, however, always needed." The term "methodology" is vague here. Serinaldi (2015) cautions against comparing results from different hazard scenarios rather than from different copulas.

Thank you for your comment, we agree with you and adjusted the manuscript accordingly replacing "methodologies" by "hazard scenarios".

The authors have provided point-wise response to my comments. However, they have not addressed several of my comments, as pointed out earlier, and some of their responses are not satisfactory. As a result, the paper's quality remained nearly unchanged, with only minor changes to the written portion. Before accepting for final publication, the following points should be addressed and I have highlighted them again:

Thank you for your review and many helpful comments to improve our study. We hope our responses given below marked in red as well as our changes in the manuscript have helped to address these issues.

1) The paper is basically addressing compound hazards and not the 'risk' as risk is a function of hazard, exposure and vulnerability. In their analysis, the last two terms were not addressed. In Abstract and elsewhere, they should replace the word with 'hazards' and not the risk. At the same time, 'risk' here is not any indirect consequences as pointed in the Abstract.

On line 12: They may clearly state, ""The compound flood hazards from high sea levels and high river discharge are often estimated using copulas".

Thank you for your comment, we agree with you and adjusted the manuscript accordingly.

2) Line 92: "Hybrid statistical-hydraulic/hydrodynamic modelling frameworks….

Ganguli et al. (2020), in a coupled statistical-hydrodynamic modelling framework showed projected changes in compound flood hazard is limited to 34% of the sites with a substantial role of SLR in modulating compound flood hazard.

Thank you for your comment, we adjusted the manuscript accordingly and added information about this in the introduction where we think it is the most relevant (lines 66-69).

3) In their response letter, they indeed acknowledge that uncertainties are linked to available/length of datasets; however, it should be highlighted in the manuscript. Further, different sources of uncertainty, for example, re-analysis driven storm surge as well as different modelling approaches, such as S-hype and E-hype as implemented here should be clearly explained in the Discussion section.

Thank you for your comment, we added clarity (lines 479-483 and 522-526) emphasizing that uncertainties are partly linked to the available/length of datasets in addition with already included sentences on this topic (lines 120-121 / 166-169 / 462-465).

4) Although it was suggested how to estimate the uncertainty of the E-hype and S-hype models using a simple ratio approach, this part was not addressed. This is indeed not an outside scope of the research when one is estimating the probabilistic hazard of compound floods using a multivariate copula approach. This is because the uncertainty from the individual modelling components, including the univariate flood hazard estimates, are also propagated in multivariate risk estimation.

Thank you for your comment and we agree, the uncertainty from the individual modelling components are are also propagated in multivariate risk estimation. This study, actually aims to quantify those uncertainties in investigating the sensitivity of the choice of data sources within the bivariate analysis. As we can notice in the figure below looking at the uncertainties within the univariate study (the upper panel shows the difference between the 95th percentile and the 5th percentile of the confidence interval from the univariate GEV fit and the lower panel displays the ratio between both), uncertainties from the GEV analysis are larger for the S-Hype model than for the E-Hype model and, as expected are larger for higher RPs as we can also expect from looking at Figure 4b.

[Figure]

5) The definition of 'AND' and 'OR' return period estimates are still not revised properly: 'AND' case corresponds to when both high sea levels and river discharge exceed the respective random variables concurrently, whereas the 'OR' case indicates when either of the extremes exceeds the respective random variable with a time offset within a limited time interval. Please refer to the Requena et al. (2013).

Thank you for your comment, we adjusted the manuscript lines (258-261).

6) The organization of this paper should be revised again: I re-emphasize that the copula sensitivity, i.e., selection of copulas on joint hazard estimation can't be the crucial point and novelty of this paper. Such studies were already been shown in the literature. Further, there is already a large body of literature that focuses on the selection of the best copula models as well as the goodness-of-fit of the copulas. The focus of this study should be insight-driven and to decipher whether it is the high sea level or the high river discharge that contributes to coastal compound floods and if it is the latter, what are the uncertainty sources that potentially mediate the river flood hazards across the Nordic countries, which eventually have the potential to shape the coastal compound flood hazard.

Thank you for your comment. We agree the selection of copulas can't be the crucial point and novelty of this paper. However, we think, as mentioned by reviewer 2, that "the novelty of this study stems from testing the sensitivity of the estimated hazard to the different data sources." This comment follows our argumentation given in the last review process to reviewer 2 and, we think, has been addressed fairly clearly within this study but has been re-emphasized in the paper (lines 86-87). The correlation analysis indeed shows that high river discharge contributes to compound coastal floods over high sea level and has been shown in the manuscript. We also think that a broader spatial study over Scandinavia would be needed to address which driver between high river discharge or high sea level contributes mainly to coastal compound floods across the Nordic countries.

7) One of the main findings of this study is that river discharge dominates over coastal surge in shaping coastal compound floods, which is interesting and is true for gauges > 60°N latitude (Ganguli and Merz, 2019), where rare occurrences of compound floods is reported in the literature due to decrease in relative sea level rise across Nordic countries due to vertical crustal movement (Weisse et al., 2021).

Thank you for your comment, we agree that this is interesting and added this to the introduction where we think it fits best (lines 53-57). However, we do not think this is the core of the paper (see previous answer to comment 6). The specific site of Halmstad is located at around 56.6°N latitude where the mean sea level is expected to rise slightly even under RCP2.6 at Ringhals and Viken stations (located North and South of Halmstad respectively (Hieronymus and Kalén, 2020).

References:

Ganguli, P. and Merz, B.: Trends in Compound Flooding in Northwestern Europe During 1901–2014, Geophysical Research Letters, 46, 10810–10820, https://doi.org/10.1029/2019GL084220, 2019.

Ganguli, P., Paprotny, D., Hasan, M., Güntner, A., and Merz, B.: Projected Changes in Compound Flood Hazard From Riverine and Coastal Floods in Northwestern Europe, Earth's Future, 8, e2020EF001752, https://doi.org/10.1029/2020EF001752, 2020.

Requena, A. I., Mediero, L., and Garrote, L.: A bivariate return period based on copulas for hydrologic dam design: accounting for reservoir routing in risk estimation, Hydrology and Earth System Sciences, 17, 3023–3038, https://doi.org/10.5194/hess-17-3023-2013, 2013.

Weisse, R., Dailidienė, I., Hünicke, B., Kahma, K., Madsen, K., Omstedt, A., Parnell, K., Schöne, T., Soomere, T., Zhang, W., and Zorita, E.: Sea level dynamics and coastal erosion in the Baltic Sea region, Earth System Dynamics, 12, 871–898, https://doi.org/10.5194/esd-12-871-2021, 2021.

Hieronymus, M. and Kalén, O.: Sea-level rise projections for Sweden based on the new IPCC special report: The ocean and cryosphere in a changing climate, Ambio, 49, 1587–1600, https://doi.org/10.1007/s13280-019-01313-8, 2020.

---

## Author Response (AR3)

Thank you for the work you have done in addressing the second round reviewers. In light of the reviews provided by the two reviewers, your reply and track changes document, I will ask Reviewer 2 for a last round of revision. On my side, I also have some minor comments:

Thank you for your review and helpful comments to improve our study. We hope our responses given below marked in red as well as our changes in the manuscript have helped to address these issues.

**Comments**

Line 15: You state before that the main objective of the article is to " investigate the influence of different data sources coming from observations and models as well as the choice of copula on extreme water level estimates". The first sentence presenting your result is discussiong a different topic. I believe that reorganizing the second part abstract would be beneficial.

Thank you for your comment, we agree it could be misunderstood and adjusted the abstract accordingly.

Figure 2: The content is fine, although the design could be improved.

Thank you for your comment, we adjusted the figure hopefully with an improved design.

Figure 3/4: Please choose between [] and () for units.

Thank you for your comment, we adjusted the figures accordingly.

General: Just replacing the work "risk" by "hazard" does not work everywhere. Example, line 271, "the hazard of flooding" would read better as "the flood hazard"

Thank you for your comment, we adjusted the manuscript accordingly.